# Sphingolipids mediate polar sorting of PIN2 through phosphoinositide consumption at the *trans*-Golgi network

Yoko Ito[1], Nicolas Esnay [1,6], Matthieu Pierre Platre [2,7], Valérie Wattelet-Boyer[1], Lise C. Noack [2], Louise Fougère[1], Wilhelm Menzel[1,8], Stéphane Claverol [3], Laetitia Fouillen [1,4], Patrick Moreau [1,5], Yvon Jaillais [2] & Yohann Boutté [1✉]

The lipid composition of organelles acts as a landmark to define membrane identity and specify subcellular function. Phosphoinositides are anionic lipids acting in protein sorting and trafficking at the *trans*-Golgi network (TGN). In animal cells, sphingolipids control the turn-over of phosphoinositides through lipid exchange mechanisms at endoplasmic reticulum/ TGN contact sites. In this study, we discover a mechanism for how sphingolipids mediate phosphoinositide homeostasis at the TGN in plant cells. Using multiple approaches, we show that a reduction of the acyl-chain length of sphingolipids results in an increased level of phosphatidylinositol-4-phosphate (PtdIns(4)P or PI4P) at the TGN but not of other lipids usually coupled to PI4P during exchange mechanisms. We show that sphingolipids mediate Phospholipase C (PLC)-driven consumption of PI4P at the TGN rather than local PI4P synthesis and that this mechanism is involved in the polar sorting of the auxin efflux carrier PIN2 at the TGN. Together, our data identify a mode of action of sphingolipids in lipid interplay at the TGN during protein sorting.

[1] Laboratoire de Biogenèse Membranaire, Univ. Bordeaux, Villenave d'Ornon, France. [2] Laboratoire Reproduction et Développement des Plantes, Université de Lyon, ENS de Lyon, UCB Lyon1, CNRS, INRAE, Lyon, France. [3] Platforme Proteome, Univ. Bordeaux, Bordeaux, France. [4] MetaboHub-Bordeaux Metabolome INRAE, Villenave d'Ornon, France. [5] Bordeaux Imaging Centre, Plant Imaging Platform, UMS 3420 University of Bordeaux-CNRS, INRAE, Villenave-d'Ornon Cedex, France. [6] Present address: BioDiscovery Institute and Department of Biological Sciences, University of North Texas, Denton, TX, USA. [7] Present address: Plant Molecular and Cellular Biology Laboratory and Integrative Biology Laboratory, Salk Institute for Biological Studies, La Jolla, CA, USA. [8] Present address: Leibniz Institute of Plant Biochemistry, Halle (Saale), Germany. ✉email: yohann.boutte@u-bordeaux.fr

Post-Golgi protein sorting is a fundamental process to direct proteins to polar domains of eukaryotic cells[1,2]. The *trans*-Golgi Network (TGN) is an essential organelle acting in cargo sorting. TGN malfunctioning results in serious diseases as well as cell polarity, differentiation, and organ development defects in both animal and plant kingdoms[2,3]. Lipid interplay between sphingolipids (SLs), sterols and phosphoinositides, is thought to orchestrate sorting and trafficking of secretory cargos at the TGN[4–6]. Sphingolipids and sterols are enriched at the TGN and are important for protein sorting and trafficking[7–9]. Importantly, sphingolipid-metabolic flux controls phosphoinositide homeostasis through phosphoinositides/sterols or phosphoinositides/phosphatidylserine (PS) exchanges between the ER and the TGN[5,6,10]. This effect of sphingolipids over phosphoinositides is crucial as phosphoinositides favor vesicle budding and fission and act in polarized trafficking in concert with small GTPases or elements of the exocyst complex[4,11–13]. Additionally, phosphoinositides recruit adaptor proteins or membrane curvature-sensitive proteins that help selecting cargos and forming vesicles[14–16]. In animal cells, the transfer of phosphocholine from phosphatidylcholine (PC) to ceramide occurs at the TGN and produces sphingomyelin and a diacylglycerol (DAG) molecule that favor negative membrane curvature and fission, and activates the PI4KinaseIIIβ (PI4KIIIβ), which locally produces PI4-phosphate (PI4P)[5,14,17]. PI4P recruits both CERT, facilitating ceramide transfer, and oxysterol-binding proteins (OSBP), which exchange PI4P for sterols at ER-TGN contact sites[6,18]. This process negatively feedbacks on OSBP localization at the TGN and the transfer of ceramide from the ER to the TGN. Hence, there is a homeostatic control of sphingolipid synthetic flow over PI4P turnover at ER/TGN contact sites, which is dependent on PI4P consumption through OSBP-mediated sterol exchange and proximity of OSBP with PI4K[5,6]. Some OSBP-related proteins (ORPs) operate a PS/PI4P exchange at the TGN rather than sterol/PI4P exchange[10]. In plants, OSBP proteins have been evidenced at the ER to *cis*-Golgi interface[19]. However, no ER/TGN membrane contact sites have been unambiguously shown so far in plants, probably due to the highly dynamic nature of the TGN which can detach from Golgi apparatus and become an independent organelle[20,21]. Nonetheless, sphingolipids and sterols are enriched at the TGN[22,23]. Moreover, PI4P and PS participate to establish an electrostatic territory at the plant TGN[24,25]. Interestingly, alteration of either the acyl-chain length of sphingolipids or PI4KIIIβ function result in swollen TGN-vesicles being less interconnected with membrane tubules and alteration in TGN-mediated sorting of the auxin efflux carrier PIN2 that localizes in a polar fashion at apical membrane of root epidermal cells, indicating a potential interplay between sphingolipids and phosphoinositides at the TGN in plant cells[22,26]. We employed a combination of live cell biology, immuno-purification of targeted TGN compartments, label-free proteomics and lipidomics approaches to now reveal that the acyl-chain length of sphingolipids plays a role in phosphoinositide homeostasis and sorting of PIN2 at the TGN, independently from sterols or PS homeostasis or from phosphoinositides-related kinases or phosphatases. Unexpectedly, we identified that phosphoinositides-specific phospholipases C (PI-PLCs) acts in sphingolipid-mediated PI4P consumption at the TGN. Moreover, our results indicate that the sphingolipids/PI4P lipid interplay plays a role in sorting of the auxin efflux carrier PIN2 at the TGN. Altogether, our results establish a mode of action of sphingolipids on phosphoinositide homeostasis during protein sorting through another mode of PI4P consumption than the PI4P/sterols exchange mechanism.

## Results

**Gravitropic defects induced by metazachlor are dependent upon PI4P synthesis at the TGN.** In *Arabidopsis* root epidermal cells, PI4P mainly resides at the plasma membrane (PM) and sparsely at the TGN/EEs[24,27]. At the TGN, local synthesis of PI4P partly occurs through the TGN-localized PI4-Kinase β1 (PI4Kβ1), which acts redundantly with PI4Kβ2[13,26,28]. To test the hypothesis of a crosstalk between sphingolipids and PI4P at the TGN we treated either wild-type or *PI4Kβ1;β2* double mutant plants with metazachlor (Mz), a chemical allowing a fine-tunable reduction of C24- and C26-Very-Long Chain Fatty Acids (VLCFAs), which are abundant in sphingolipids[22]. As phenotypic readout assay we looked at the ability of the root to reorient its growth direction following a gravistimulus, i.e., root gravitropism, given that Mz alters root gravitropism in a PIN2-dependent manner[22]. As published before, we found that Mz alters root gravitropism of wild-type seedlings at 50 nM, and even more strongly at 100 nM (Fig. 1a)[22]. Contrastingly, *PI4Kβ1;β2* double mutant seedlings are comparatively less sensitive to Mz even at 100 nM (Fig. 1b). This difference of sensitivity between the wild-type and the *PI4Kβ1;β2* double mutant indicates that the phenotypic effect of Mz is mediated, at least in part, through the pool of PI4P at the TGN.

**The quantity of PI4P within the TGN is modulated by very long chain fatty acids.** Next, we checked the localization and relative quantity of PI4P at the TGN. To this end, we used a library of complementary genetically encoded fluorescent biosensors specific of phosphoinositides[24,25,27]. Fluorescence of the PI4P biosensor 1× PH domain of the Human FAPP1 protein fused to

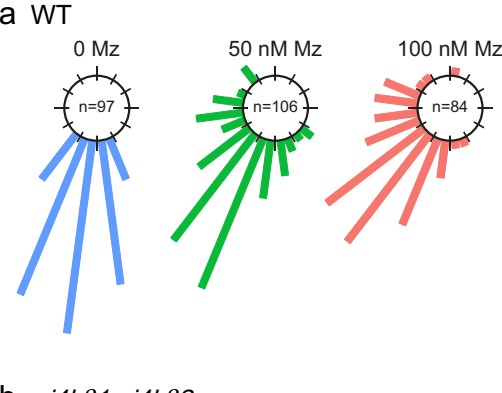

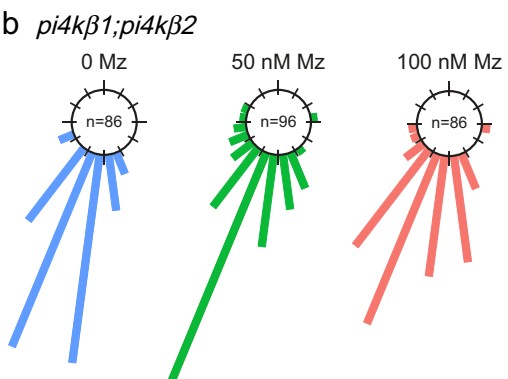

**Fig. 1 Metazachlor-induced root gravitropism defects are dependent upon PI4P synthesis at TGN.** Root gravitropism assay of wild-type (**a**) and *pi4kβ1;pi4kβ2* double mutant (**b**), untreated (0 Mz) or treated with 50 nM metazachlor (Mz) or 100 nM Mz. After gravistimulation, root angles are calculated, distributed in 30° angle classes and plotted in circular charts where each bars correspond to the effective of roots in each classes of angles, from 0° (bottom of the chart) to 180° (top of the chart). Thus, the direction of the bars indicates the direction of growth after gravistimulation. The *pi4kβ1;pi4k β2* double mutant is more insensitive to Mz treatment than wild-type. n values are indicated in the graphs.

mCITRINE (mCIT-1×PH$^{FAPP1}$)[27] was increased at intracellular compartments when the roots were treated with either 50 nM or 100 nM Mz as compared to untreated roots (Fig. 2a, b. All $P$-values are described in Supplementary Data 1). The PH domain of the FAPP1 protein binds to both PI4P and the small GTPase ARF1, which partly localizes at the TGN[29,30]. Hence, we checked that the increase of signal observed upon Mz was not due to increased binding of the mCIT-1×PH$^{FAPP1}$ sensor to the ARF1 protein. We used the mCIT-1×PH$^{FAPP1-E50A}$ and mCIT-1×PH$^{FAPP1-E50A-H54A}$ sensors mutated for ARF1 binding but not PI4P binding[24]. Consistent with previous observations in *Nicotiana benthamiana*[24], we could hardly detect any signal at the TGN in both mCIT-1×PH$^{FAPP1-E50A}$ and mCIT-1×PH$^{FAPP1-E50A-H54A}$ untreated roots cells (Fig. 2d and Supplementary Fig. 1a). In contrast, the signal of both biosensors was clearly increased in intracellular compartments in Mz-treated cells (Fig. 2d, e and Supplementary Fig. 1a, b). In

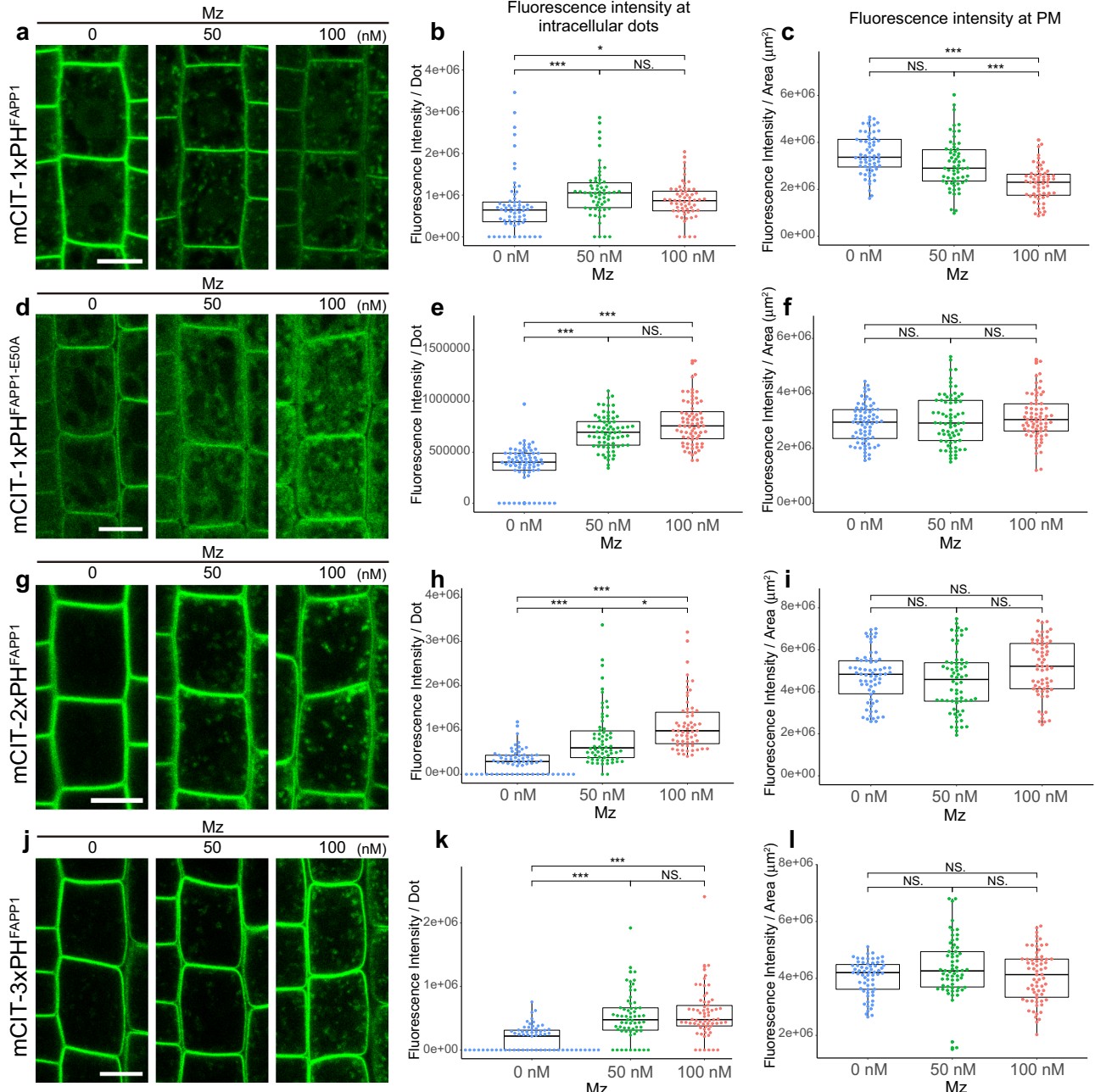

**Fig. 2 Very long chain fatty acids are required for the intracellular distribution of PI4P.** Confocal micrographs of *Arabidopsis* root epidermal cells expressing either the PI4P biosensor mCIT-1×PH$^{FAPP1}$ (**a**), mCIT-1×PH$^{FAPP1-E50A}$ (**d**), mCIT-2×PH$^{FAPP1}$ (**g**), mCIT-3×PH$^{FAPP1}$ (**j**) upon 0, 50, or 100 nM Mz (Mz) treatment. Fluorescence intensity at intracellular dots (**b**, **e**, **h**, **k**) and at PM (**c**, **f**, **i**, **l**) was quantified. All the PI4P biosensors tested showed significant increase of the signal intensity at intracellular dots upon metazachlor (Mz) treatment while only mCIT-1×PH$^{FAPP1}$ showed a decrease of the signal at PM (**a**–**l**). (**b**, **c**) $n = 63$ cells from 21 roots (0 nM Mz) and 60 cells from 20 roots (50 and 100 nM). **e**, **f** $n = 72$ cells from 24 roots for each treatment. **h**, **i** $n = 66$ cells from 22 roots (0 and 100 nM), 63 cells from 21 roots (50 nM). **k**, **l** $n = 63$ cells from 21 roots (0 and 100 nM), 66 cells from 22 roots (50 nM). Statistics were done by two-sided Dwass-Steel-Critchlow-Flinger multiple comparison test with Monte Carlo method (10,000 iterations), *$P < 0.01$, ***$P < 0.0001$. Each element of the boxplot indicates the following value: center line, median; box limits, the first and third quartiles; whiskers, 1.5× interquartile range; points above or below the whiskers, outliers. Scale bars, 10 μm.

addition, the intensity of ARF1-GFP in dots was not altered upon Mz (Supplementary Fig. 1d, e). Hence, we excluded the possibility that the increase of fluorescence intensity of the mCIT-1×PH$^{FAPP1}$ PI4P sensor was due to higher binding to ARF1 or to an increased level of ARF1 at the TGN upon Mz treatment. To reinforce our data, we analyzed PI4P sensor lines with higher number of PH domains, i.e., mCIT-2×PH$^{FAPP1}$ and mCIT-3×PH$^{FAPP1}$, which increase the avidity of the sensor to PI4P[24]. In untreated roots, both mCIT-2×PH$^{FAPP1}$ and mCIT-3×PH$^{FAPP1}$ sensors displayed strong PM labeling and almost undetectable levels of fluorescence at the TGN[24,27] (Fig. 2g, j). Strikingly, both mCIT-2×PH$^{FAPP1}$ and mCIT-3×PH$^{FAPP1}$ showed clear increase of signal intensity in intracellular compartments upon Mz (Fig. 2h, k). We also checked the fluorescence intensity of PI4P sensors at the PM and observed that mCIT-1×PH$^{FAPP1}$ intensity at the PM was decreased upon Mz (Fig. 2c). Contrastingly, mCIT-2×PH$^{FAPP1}$ and mCIT-3×PH$^{FAPP1}$ intensity were not decreased at the PM upon Mz (Fig. 2i, l). These results are consistent with the previous observation that mCIT-2×PH$^{FAPP1}$ and mCIT-3×PH$^{FAPP1}$ dwell time is higher than mCIT-1×PH$^{FAPP1}$ in PI4P-riched membranes[24]. Similarly, mCIT-1×PH$^{FAPP1-E50A}$ and mCIT-1×PH$^{FAPP1-E50A-H54A}$ intensities at the PM were not decreased upon Mz (Fig. 2f and Supplementary Fig. 1c). To confirm our observation in mCIT-1×PH$^{FAPP1}$ sensor line, we analyzed another PI4P biosensor that strictly labels the pool of PI4P at the PM i.e., the P4M domain of the *Legionella pneumophila* SidM protein fused to mCITRINE[24] (mCIT-P4M$^{SidM}$). Our results were similar to mCIT-1×PH$^{FAPP1}$ sensor, the intensity at the PM was decreased upon Mz treatment (Supplementary Fig. 1f, g). Hence, the effect of Mz on the pool of PI4P at the PM appears to be slighter as compared to the consistent increase we observed at the TGN. Together, our analyses show that Mz consistently promotes the localization of PI4P sensors at the TGN, raising the possibility that the PI4P level within this compartment are indeed upregulated by this treatment.

To confirm our biosensor approach, we directly quantified the level of phosphoinositol monophosphate (PIP) in immuno-purified (IP) SYP61-SVs/TGN compartments from control and Mz-treated seedlings (Fig. 3a). We first checked the efficiency of our immuno-purification by loading equal amount of IP input and output fractions on SDS-PAGE (Supplementary Fig. 2a) and performed western blotting with an anti-GFP antibody. Our results showed that the IP output fraction was enriched for SYP61-CFP as compared to the IP input fraction (Supplementary Fig. 2b). Moreover, the SVs/TGN-resident protein ECHIDNA was also found to be enriched in the IP output fraction (Supplementary Fig. 2c), confirming the efficient purification of SYP61-SVs/TGN compartments. We next established a LC-MS/MS pipeline that quantifies total phosphatidylinositol monophosphate (PIP) levels from these IP fraction. As such, we measured the total levels of PIPs (i.e., the sum of PI3P, PI4P and PI5P), keeping in mind that PI4P represents ~80% of total phosphatidylinositol monophosphate in cells[31]. Using this method, we detected an increased level of phosphatidylinositol monophosphate (1.8-fold increase) in immuno-purified fractions of TGNs treated with Mz as compared to the control fractions (Fig. 3b, c). These results are fully consistent with the increased fluorescence level of PI4P biosensors at the TGN observed after Mz treatment. Taken together, direct measurements and our sensor approach indicate that Mz treatment increases the quantity of PI4P at the TGN, and they suggest that the acyl-chain length of lipids is critical in regulating PI4P subcellular accumulation.

**Very long chain fatty acids impact PI4P accumulation at the TGN.** Mz directly targets the 3-ketoacyl-coenzyme A synthase KCS2, KCS20, and KCS9 enzymes which condense a C2 moiety

of a malonyl-CoA on a C22 acyl-CoA to produce C24- and C26-VLCFAs, including α-hydroxylated h24- and h26-VLCFAs and non α-hydroxylated 24- and 26-VLCFAs[22,32–34]. To confirm that Mz alters VLCFAs composition at TGN, we characterized the fatty acid composition by GC-MS of the SYP61 IP compartment in untreated and Mz-treated seedlings. Our results revealed that, in SVs/TGN IPs, Mz strongly decreases h24- and h26-VLCFAs (4.7-fold decrease), which is specific to SLs, as well as 24- and 26-VLCFAs (2.5-fold decrease) (Fig. 3d, e). Moreover, Mz neither altered the level of non α-hydroxylated 16- and 18-fatty acids, which are a hallmark for phospholipids, nor it altered the total level of C20, C22 fatty acids or sterols in SYP61-SVs/TGN IP compartments (Fig. 3d, e and Supplementary Fig. 2d, e). Thus, we confirmed that Mz specifically alters VLCFAs at TGN. Moreover, we could observe a 2.3-fold enrichment of h24- and h26-VLCFAs as compared to 24- and 26-VLCFAs in SYP61-SVs/TGN IP compartments (Fig. 3f). These results are consistent with the previous observation that h24 and h26 are enriched at TGN[22]. As our results indicate that the quantity of h24 is 10 times higher that h26, we tested whether the reduction of h24 was causal to the phenotype induced by Mz (i.e., accumulation of PI4P at the TGN) by performing rescue experiments in which we incubated Mz-treated seedlings with h24:0 fatty acids. First, we verified that h24:0 in Mz-treated seedlings was coming back to the level of untreated seedlings (Fig. 4a, b). We then quantified the fluorescence of the mCIT-2×PH$^{FAPP1}$ PI4P biosensor in these seedlings and found that h24:0 rescued, at least partially, the Mz-induced increase of PI4P at intracellular dots (Fig. 4c, d). This result suggests that the PI4P accumulation at the TGN observed upon Mz treatment is indeed due to a reduction in h24:0.

Next, we verified whether KCS enzymes are the target of Mz for the induction of PI4P at the TGN. Using root gravitropism assay as a phenotypic readout of PIN2 polarity we previously showed that both the *kcs2;20* double mutant and the *kcs9* single mutant are hypersensitive to Mz[22]. As KCS2, KCS20, and KCS9 act redundantly in C22 to C24 fatty acid elongation (both α-hydroxylated and non-α-hydroxylated), we analyzed the localization of the mCIT-3xPH$^{FAPP1}$ PI4P biosensor into *kcs9* single mutant treated with Mz in a dose-response assay. Our results show a slight but significant increase of PI4P in intracellular compartments in *kcs9* mutant seedlings treated with 10 nM Mz but not in wild-type seedlings treated at the same concentration (Supplementary Fig. 3a, b). These results confirmed that VLCFAs are involved in PI4P homeostasis at the TGN. However, given the genetic redundancy between KCSs, we observed that the increase of PI4P in intracellular dots was similar between wild-type and *kcs9* mutant at higher Mz concentration (Supplementary Fig. 3b).

Taken together, lipid analyses of immuno-purified TGN, add-back experiments and genetic approach confirm that VLCFA are critical for PI4P homeostasis at the plant TGN, and suggest a possible crosstalk between sphingolipid and phosphoinositides subcellular patterning.

**The acyl-chain length of sphingolipids, not of phospholipids, are involved in PI4P homeostasis at the TGN.** VLCFAs constitute about 85% and 2% of the pools of sphingolipids and phospholipids in *Arabidopsis* root, respectively[22]. Although minor, the 2% of VLCFAs in the pool of phospholipids could mediate the effect of Mz on PI4P increase at TGN. PS is by far the major phospholipid with VLCFA in plants[25,35]. Moreover, PS acts in concert with PI4P to generate an electrostatic gradient between the PM and the TGN[25]. Hence, we wondered whether the increase of PI4P upon Mz would also be visible for PS. Our results did not evidence obvious changes in the fluorescence intensity of the PS biosensor C2 domain of bovine Lactadherin fused to

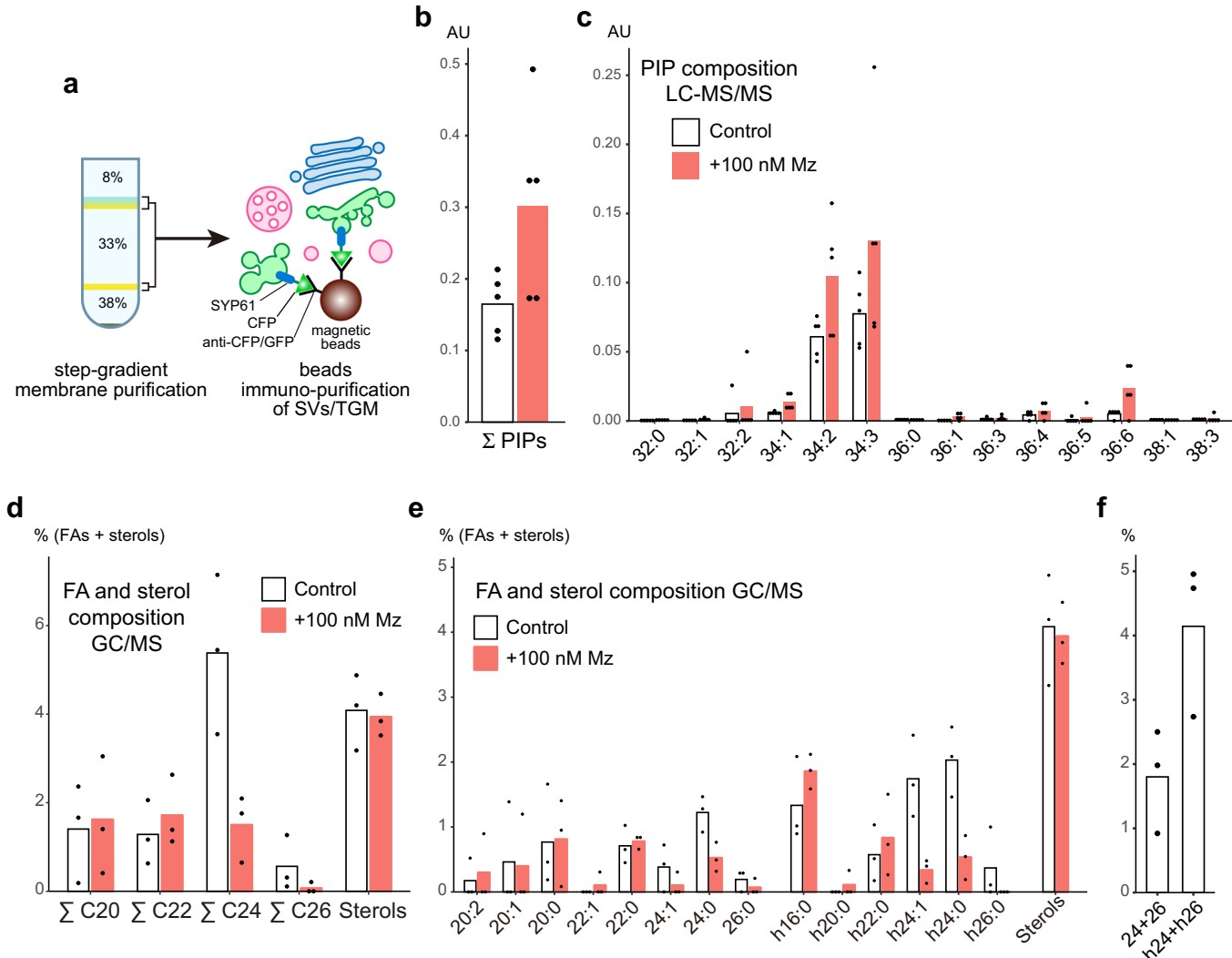

**Fig. 3 Composition of phosphoinositol monophosphate and fatty acids at TGN. a** Immuno-purification of intact TGN compartments. Step-gradient-purified membrane fraction was incubated with magnetic beads conjugated with CFP/GFP antibodies, and the intact compartments labeled by the SVs/TGN marker SYP61-CFP were immuno-purified. **b, c** LC-MS/MS analysis of phosphoinositol monophosphates (PIPs) in immuno-purified SYP61-SVs/TGN fraction in control condition (white) or treated with 100 nM metazachlor (Mz) (red) (*n* = 5 biological replicates, the dots show the dispersion of data, AU Arbitrary Unit (area of peak compound/area of internal standard)). **b** Mz increases the total amount of PIPs. **c** Detailed analysis shows that all species of PIPs are increased upon Mz. **d, e** GC-MS analysis of fatty acid and sterol composition of immuno-purified SYP61-SVs/TGN fraction in control condition (white) or treated with 100 nM Mz (red) (*n* = 3 biological replicates, the dots show the dispersion of data). **d** While Mz strongly decreased the sum of C24 and the sum of C26 fatty acids it did not affect the sterol content of SVs/TGN. **e** Detailed analysis shows that both non-α-hydroxylated 24:0, 24:1, 26:0 and α-hydroxylated h24:0, h24:1, h26:0 are reduced at TGN upon Mz. **f** In untreated control TGN, h24, and h26 (specific of sphingolipids) are two-fold enriched as compared to non-α-hydroxylated 24 and 26 fatty acids (present in both sphingolipids and PS).

mCITRINE (mCIT-C2^LACT)[25] at the PM or in intracellular compartments upon Mz treatment (Fig. 5a–c). Furthermore, contrastingly to what we observed upon Mz treatment, we did not observe any accumulation of the mCIT-2×PH^FAPP1 PI4P marker at the TGN in the *phosphatidylserine synthase1* (*pss1*) mutant, which does not produce any PS[25] (Fig. 5d, e). Thus, PS is not a major regulator of PI4P accumulation at the TGN. Our results further suggest that the increase of PI4P at the TGN upon Mz is unlikely to be correlated with a PS/PI4P exchange mechanism.

Next, we took advantage from the fact that α-hydroxylated fatty acids constitute a specific signature of sphingolipids and are not incorporated at all into phospholipids[22,36]. We thus performed add-back experiment as previously described but this time with non α-hydroxylated 24:0 fatty acid. We confirmed that the amount of 24:0 in Mz-treated seedlings was coming back to the level of untreated seedlings (Supplementary Fig. 4a, b). By

contrast to h24:0, which could rescue the localization of PI4P in Mz-treated plants, exogenous treatment with 24:0 had no impact on PI4P localization and was unable to counteract the effect of Mz (Supplementary Fig. 4c, d). As h24:0 is specific for sphingolipids, our results suggest that sphingolipids act on PI4P homeostasis at the TGN and further exclude a role of other 24:0-containing lipids such as PS.

SLs display a wide structural diversity and contain several subclasses of lipids, including free Long Chain Bases (LCBs) that can be amidified with VLCFAs to produce VLCFA-ceramides on which an inositolphosphate group is added to generate inositolphosphorylceramide (IPC)[37]. IPC is further glycosylated to produce glycosyl-inositolphosphorylceramide (GIPC), the most abundant form of SLs in plants[36,37] (Fig. 6a). To further confirm that the increase of PI4P at the TGN is due to VLCFA-containing SLs, we decided to use the characterized ceramide

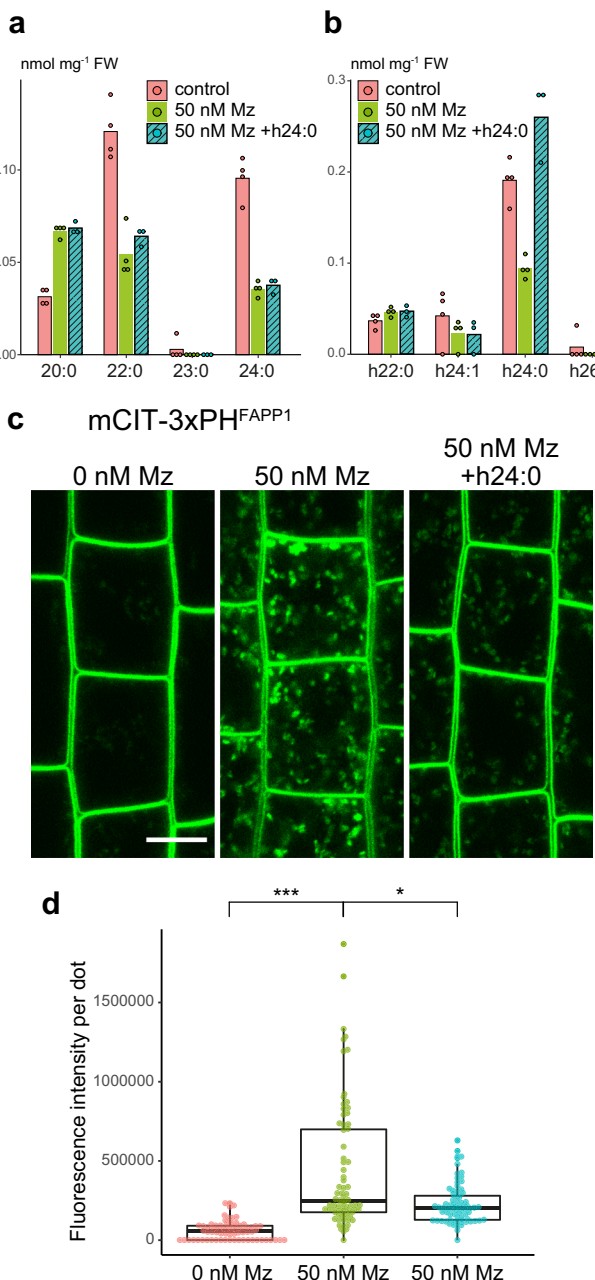

**Fig. 4 α-hydroxylated h24:0 rescues the metazachlor-induced PI4P distribution defect at TGN. a, b** GC-MS analysis of *Arabidopsis* roots in control condition (pink), metazaclor (Mz)-treated roots (green), and metazachlor-treated roots implemented with h24:0 (blue) (*n* = 4 biological replicates, the dots show the dispersion of data). **a** Quantification of non-α-hydroxylated fatty acids and **b** α-hydroxylated fatty acids. External application of h24:0 in Mz-treated seedlings restores the level of h24:0 of control seedlings. **c** Confocal micrographs of *Arabidopsis* root epidermal cells expressing the PI4P biosensor mCIT-3xPH^FAPP1 upon 0, 50 nM Mz treatment or 50 nM Mz treatment + h24:0 add-back. Scale bar, 10 μm. **d** Quantification of the fluorescence intensity at intracellular dots. While Mz treatment induces PI4P accumulation in intracellular dots, external application of h24:0 rescues this defect. *n* = 75 cells from 25 roots for each condition. Statistics were done by two-sided Dwass-Steel-Critchlow-Flinger multiple comparison test with Monte Carlo method (10000 iterations). *$P < 0.01$, ***$P < 0.0001$. Each element of the boxplot indicates the following value: center line, median; box limits, the first and third quartiles; whiskers, 1.5× interquartile range; points above or below the whiskers, outliers.

synthase inhibitor fumonisinB1 (FB1). FB1 treatment does not modify the global quantity of ceramides but rather alters their composition by decreasing the quantity of VLCFA-ceramides and increasing the quantity of C16-ceramides[38] (Fig. 6b and Supplementary Fig. 5a). Similarly, the quantity of either C16-glucosylceramide (GluCer) or C16-GIPC species was increased in FB1-treated roots (Supplementary Fig. 5d, g). Our results show that FB1 treatment triggered a small but significant accumulation of mCIT-2×PH^FAPP1 PI4P biosensor in intracellular dots (Supplementary Fig. 6a, b), confirming by yet another independent approach, the importance of sphingolipids in PI4P homeostasis at the TGN. However, we also noticed that FB1 treatment had a much weaker effect as compared to Mz. This may indicate that VLCFA-ceramides have a rather small effect on PI4P subcellular distribution, which could be mediated instead by the glycosylated forms of sphingolipids, such as VLCFA-GIPCs.

**Local sphingolipid synthesis by IPCS1/2 in the Secretory Vesicle subdomain of the TGN is acting on PI4P homeostasis.** Alteration of GIPC level by genetic mean is a thorny problem as complete knockout mutants are lethal and knockdown could display normal growth with only 10% of GIPC left[39,40]. We chose to use another strategy by producing an inducible artificial microRNA (amiRNA) *Arabidopsis* line targeting both *IPC SYNTHASE 1* (*IPCS1*) and *IPCS2* genes at the same time. We evaluated the level of *IPCS1* and *IPCS2* transcript in the *IPCS1;2^amiRNA* line by quantitative RT-PCR (qRT-PCR) and could detect a 35% and 74% decreased RNA level of *IPCS1* and *IPCS2*, respectively, in the *IPCS1;2^amiRNA* line treated with the inducer β-estradiol as compared to the nontreated condition (Supplementary Fig. 7a. All primers are described in Supplementary Data 2). Consistently, the root length was decreased in the *IPCS1;2^amiRNA* line treated with β-estradiol as compared to the nontreated condition (Supplementary Fig. 7b, c). To identify the exact sphingolipid composition in the *IPCS1;2^amiRNA* line, we performed LC-MS/MS on extracts of nontreated and β-estradiol-treated *IPCS1;2^amiRNA* roots. Our results revealed that the quantities of both VLCFA-GIPC and VLCFA-GluCer species were decreased in β-estradiol-induced *IPCS1;2^amiRNA* roots, as compared to non-induced, while the quantity of ceramides species was overall increased (Fig. 6b, c, d and Supplementary Fig. 5b, e, h). Importantly, β-estradiol neither modified the quantity of ceramides, GluCer or GIPC nor it altered the composition within these pools of sphingolipids (Fig. 6b, c, d and Supplementary Fig. 5c, f, i). To address whether GIPC synthesis by IPCS impacts PI4P homeostasis we quantified the fluorescence of the mCIT-3xPH^FAPP1 PI4P biosensor in the roots of the *IPCS1;2^amiRNA* line. β-estradiol treatment did not induce the intracellular accumulation of PI4P in the wild-type. However, we observed a significant increase in PI4P sensor localization at the TGN in the *IPCS1;2^amiRNA* line upon β-estradiol induction (Fig. 7a, b). Moreover, colocalization analyses with the SVs/TGN markers VHA-a1-mRFP or ECHIDNA showed high level of colocalization (Fig. 7c, e).

Next, we wondered in which subcellular compartment IPCS enzymes are acting in sphingolipid synthesis. While the synthesis of ceramide occurs in the ER, it was previously reported that IPCS2 may localize at SYP61-SVs/TGN compartments[41]. Subcompartmentalization of SLs biosynthetic enzymes might be a way to assign distinct SLs species at defined compartments. Thus, we constructed an IPCS2-tagRFP fluorescent construct driven by native IPCS2 promoter. To check the functionality of this construct we generated a double T-DNA insertion knockout mutant of *ipcs1;ipcs2* that displays a severe developmental phenotype and seedling lethality (Supplementary Fig. 7d). The

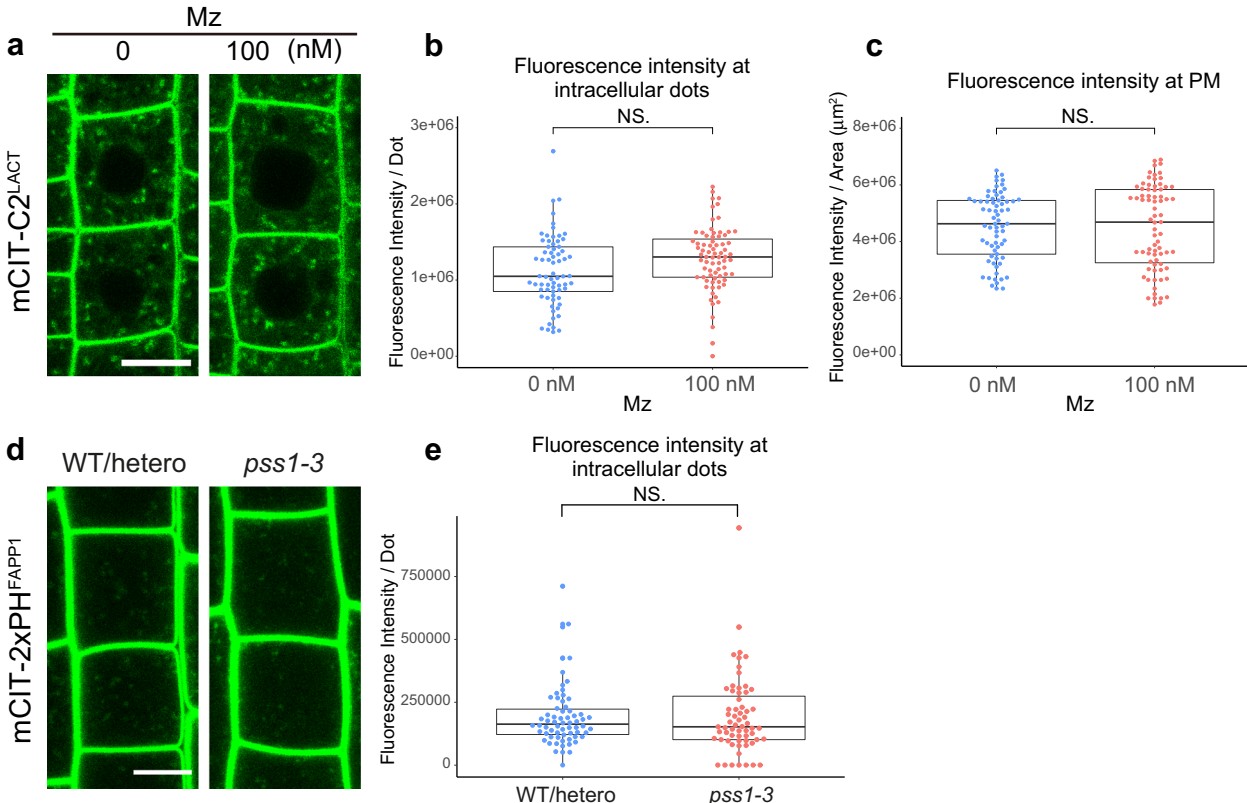

**Fig. 5 PS does not impact PI4P distribution at TGN. a** Confocal micrographs of *Arabidopsis* root epidermal cells expressing the PS biosensor mCIT-C2^LACT upon 0 or 100 nM metazachlor (Mz) treatment. Fluorescence intensity at intracellular dots (**b**) and at PM (**c**) was quantified. The distribution of the PS biosensor was not altered upon Mz treatment (**a–c**). $n = 69$ cells from 23 roots (0 nM), 75 cells from 25 roots (100 nM). **d** Confocal micrographs of *Arabidopsis* root epidermal cells expressing the PI4P biosensor mCIT-2×PH^FAPP1 in wild-type/heterozygote plants or *pss1-3* homozygote mutant. **e** Fluorescence intensity at intracellular dots was quantified. PI4P quantity at intracellular dots was not modified in *pss1-3* mutant. $n = 66$ cells from 22 roots (WT/hetero), 63 cells from 21 roots (*pss1-3*). Statistics were done by two-sided Wilcoxson's rank-sum test. Each element of the boxplot indicates the following value: center line, median; box limits, the first and third quartiles; whiskers, 1.5× interquartile range; points above or below the whiskers, outliers. Scale bars, 10 μm.

*pIPCS2::IPCS2-tagRFP* construct was able to rescue the seedling lethality phenotype indicating that IPCS2-tagRFP is functional (Supplementary Fig. 7e, f). IPCS2-tagRFP localized to intracellular compartments that strongly colocalized with ECHIDNA, a marker of the secretory vesicle subdomain of the TGN. By contrast, IPCS2-tagRFP only weakly colocalized with MEMBRIN11, a marker of the Golgi, and with Clathrin Heavy Chain a marker of the Clathrin-Coated Vesicles subdomain of the TGN (Fig. 7d, e). Thus, our results indicate that the SVs subdomain of the TGN is a main place for GIPC synthesis and suggest that local sphingolipid synthesis at SVs/TGN is acting in PI4P homeostasis.

**Proteomics of immuno-purified TGN identified PI-PLCs as potential actors of sphingolipid-mediated PI4P homeostasis.** In the aim to identify the mechanism through which sphingolipids act on PI4P level at the TGN, we performed label-free quantitative proteomics by LC-MS/MS on four biological replicates of SYP61-SVs/TGN IP compartments from control and Mz-treated seedlings (Fig. 8a). We only kept proteins that were consistently identified in each of the four biological replicates, resulting in a list of 4458 proteins. Due to the genetic redundancy in some protein families, some peptides could correspond to either one accession or the sum of several ones (indicated in Supplementary Data 3). Although the label-free methodology aims at comparing protein abundancies of a sample upon different conditions, general observations can be drawn from individual protein abundancies. Indeed, based on mass spectrometry abundance

measurement, SYP61 was one of the most abundant protein, confirming the efficiency of the IP (Supplementary Data 3). We checked the abundance of known SVs/TGN proteins and found high abundance of the V-ATPase VHA-a1[42], the RAB-GTPase-interacting YIP4a/b proteins and its interacting partner ECHIDNA[43], small GTPases of the ARF1 family, the vacuolar protein sorting45 VPS45 and its interacting proteins SYP43, SYP42, and VTI12[44,45] (Fig. 8a, Supplementary Data 4). SYP41 was found at much lower abundance than SYP43 or SYP42 indicating that SYP41 localizes at a distinct subdomain of TGN than SVs/TGN, consistently with previous work[44,45] (Fig. 8a, Supplementary Data 4). Small GTPases of the RAB family, RAB-A4b and RAB-A5d, described to localize at SVs[26], were present (Fig. 8a, Supplementary Data 4). We could not find Golgi proteins MEMBRIN11/12, SYP32, and GOT1, confirming the purity of SVs/TGN fraction over the Golgi (Fig. 8a, Supplementary Data 3). We checked the MVB/LE markers RAB-F2b, RAB-F2a, and RAB-F1 and found them in low abundance with RAB-F2b being the most present consistently to its dual MVB and TGN localization described previously[46] (Fig. 8a, Supplementary Data 4). Interestingly, none of these proteins were significantly down- or upregulated at SVs/TGN upon Mz treatment (Fig. 8b, Supplementary Data 4). Next we checked sphingolipid-related enzymes and found low abundance of the sphingolipid desaturase SLD1 and intermediate abundance of the sphingolipid kinases SPHK1/2 and LCKB2, the alkaline and neutral ceramidases ACER and NCER1/2/3, the IPC Synthase2 (IPCS2) that grafts

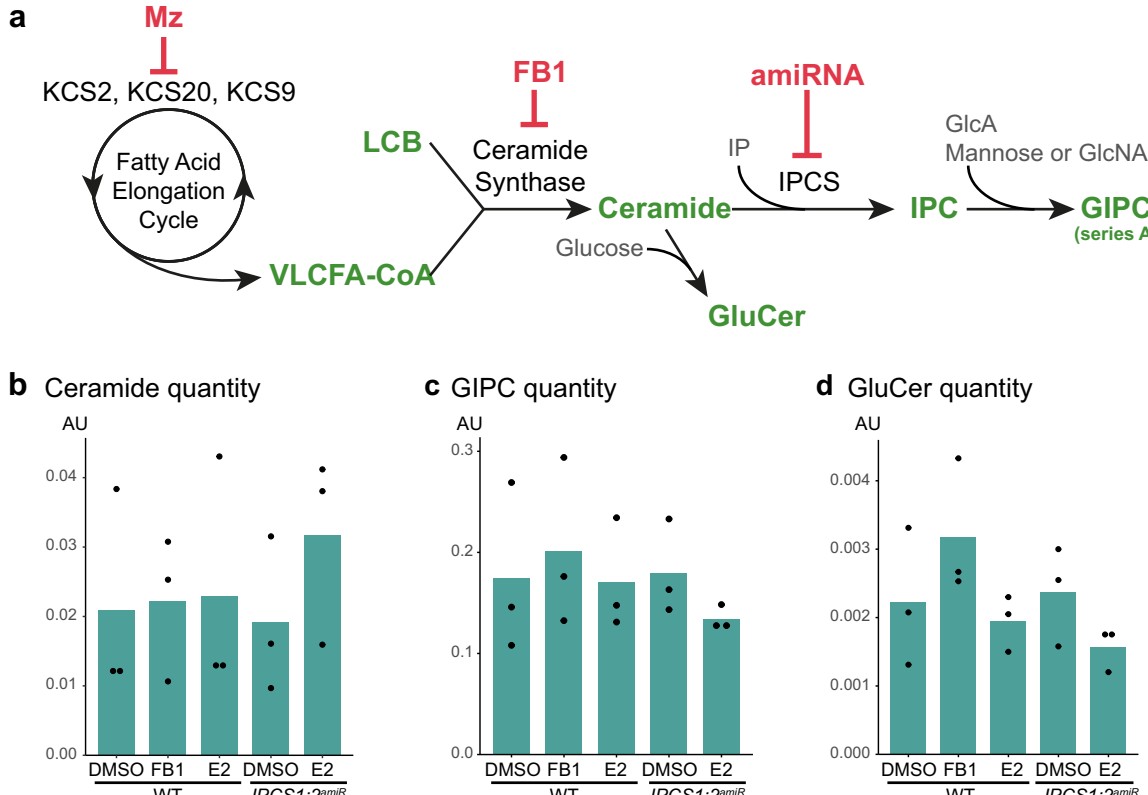

**Fig. 6 Sphingolipidomics of IPCS1;2amiRNA line and FB1-treated Arabidopsis roots. a** Simplified outline of sphingolipid biosynthesis pathway in plants. KCSs enzymes are part of the elongase complex that elongates fatty acids of 2 carbons per cycle to produce VLCFAs-CoA. Metazachlor inhibits KCS2, 20 and 9. VLCFAs-CoA is condensed with a long chain base (LCB) by the ceramide synthases LOH1 and LOH3 to produce VLCFAs-ceramide. FB1 inhibits LOH1 and LOH3. Ceramide is the substrate of the glucosylceramide synthase1 (GCS1) enzyme that grafts a glucose on ceramide to produce glucosylceramide (GluCer). Alternatively, ceramide is the substrate of the inositolphosphorylceramide (IPC) synthases (IPCSs) enzymes that graft a inositolphosphate (IP) group on ceramide to produce IPC. IPC is further glycosylated by addition of a glucuronic acid (GlcA) and mannose or to a minor extend N-acetylglucosamine (GlcNAc) residues to produce glycosylinositolphosphorylceramides (GIPCs) of series A, the most abundant form of sphingolipids in *Arabidopsis*. **b**–**d** Sphingolipidomics of *Arabidopsis* roots, these analyses are displayed in details in supplementary Fig. 5 ($n = 3$ biological replicates, the dots show the dispersion of data, AU: Arbitrary Unit (area of peak compound/area of internal standard per mg fresh weight)). **b** The total quantity of ceramides increases in the *IPCS1;2amiRNA* line induced by β-estradiol (E2) while it is not altered upon FB1 treatment. **c, d** The total quantity of GIPC (**c**) or GluCer (**d**) decreases in the *IPCS1;2amiRNA* line induced by β-estradiol (E2) while it increases upon FB1 treatment.

inositolphosphate group on ceramide to produce IPC[41] (Supplementary Fig. 8). IPCS2 abundance in SYP61/SVs IPs confirmed our previous results on IPCS2-tagRFP localization at SVs/TGN (Fig. 7d, e). We additionally found high abundance of the inositolphosphorylceramide glucuronosyltransferase1 (IPUT1) enzyme that grafts a glucuronic acid (GlcA) on IPC[47] to produce GlcA-GIPCs, and intermediate abundance of the GIPC mannosyltransferase1 (GMT1) enzyme that grafts mannose on GlcA-GIPC[48] to produce final GIPC (Supplementary Fig. 8a, Supplementary Data 4). However, none of these sphingolipid-related proteins were modified at SVs upon Mz treatment (Supplementary Fig. 8c, Supplementary Data 4). We did not detect any OSBP proteins or OSBP-related proteins (ORP) in agreement with the result that Mz did not affect sterols. We further checked PS-related enzymes and found low abundance of the PS synthase1 (PSS1), intermediate abundance of two PS flippases (ALA1 and ALA2), one PS decarboxylase (PSD3) and high abundance of two PS/PE-specific phospholipase D (PLD-gamma1/2/3 and PLD-beta1/2) (Supplementary Fig. 8b, e, Supplementary Data 4). None of these PS-related proteins were modified at SVs upon Mz treatment (Supplementary Fig. 8c). Finally, we checked the abundance of phosphoinositide-related proteins and found low

abundance of the PI synthases PIS1/2, the PI4-Kinase PI4α1, the PI(3,5)P$_2$ phosphatases SAC1-5, the PI4P phosphatase SAC8, the PI(4,5)P$_2$ phosphatases IP5P1 and SAC9, the PI3P and PI(3,5)P$_2$ phosphatases PTEN2A and PTEN2B, and high abundance of the PI4P phosphatases SAC6/7 (Fig. 8c, Supplementary Data 4). SAC6 is specific to flowers, as we performed the proteomics on seedlings we believe that SAC7 is the most abundant at SVs/TGN which is consistent with previous observation that SAC7/RHD4 localizes at post-Golgi structures in *Arabidopsis* root hair cells[49]. Importantly, none of these proteins were modified upon Mz treatment indicating that SLs do not target the localization of phosphoinositide-related phosphatases, in particular the PI4P phosphatases SAC6/7 that are the most present at SVs/TGN (Fig. 8e, Supplementary Data 4). Interestingly, we found proteins of the phosphoinositide-specific phospholipase C family (PI-PLC) that hydrolyze PIPs to produce DAG and inositol polyphosphate[50]. We found two low abundant PI-PLCs-X domain-containing proteins and high abundance of PI-PLC2/7 proteins (Fig. 8d). Both PI-PLCs-X and PI-PLC2/7 were strongly reduced at SVs/TGN upon Mz treatment (Fig. 8e). It was shown previously that PLCs hydrolyze equally well both PI4P and PI(4,5)P$_2$ in vitro and that genetic depletion of *PLCs* results in

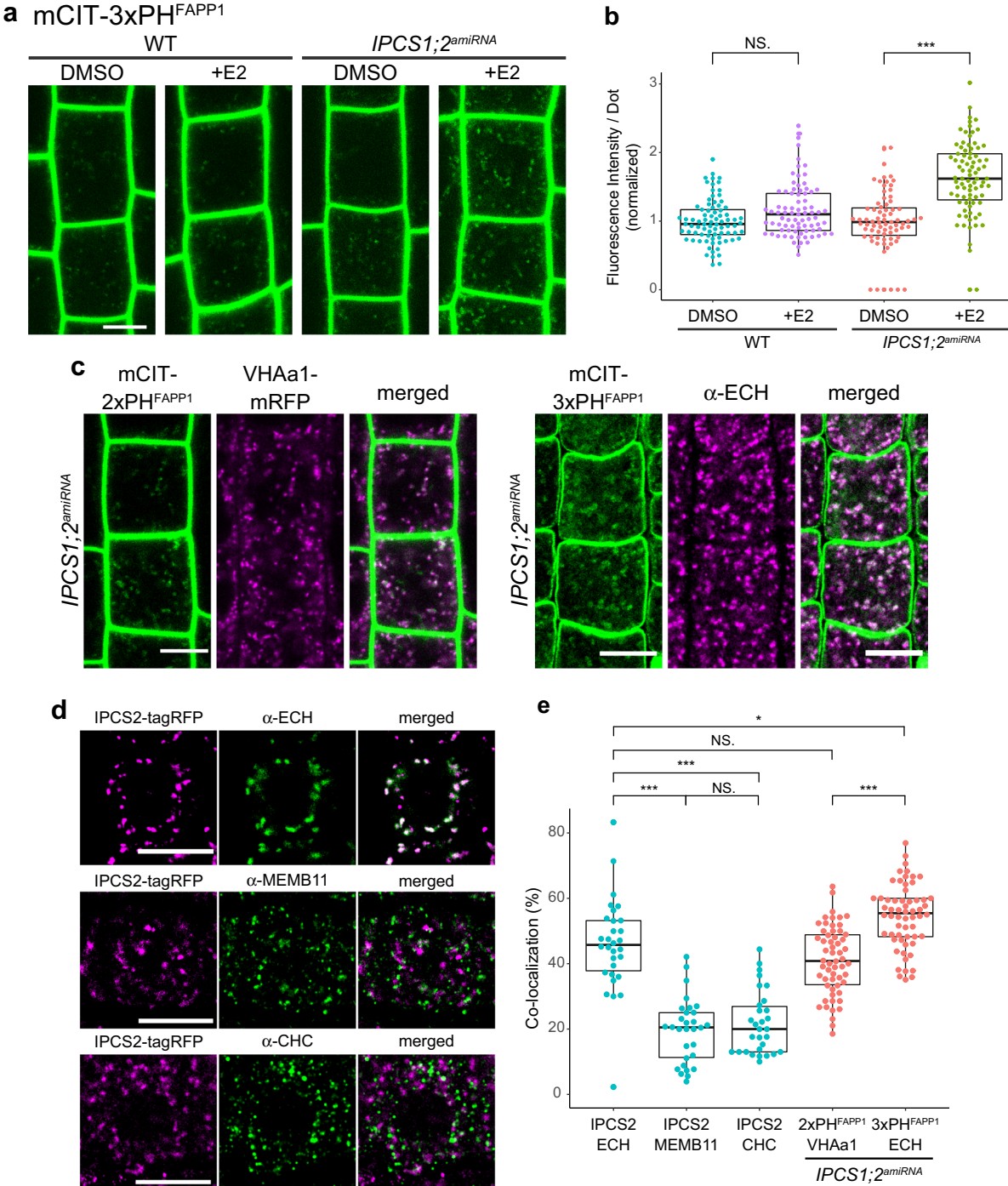

**Fig. 7 IPCS enzymes are involved in PI4P distribution at SVs subdomain of TGN where they localize. a** Confocal micrographs of *Arabidopsis* root epidermal cells expressing the PI4P biosensor mCIT-3×PH^FAPP1 in wild-type seedlings or *IPCS1;2^amiRNA* line in control condition (DMSO) or treated with the β-estradiol inducer (E2). **b** Fluorescence intensity at intracellular dots was quantified. The *IPCS1;2^amiRNA* line induced by β-estradiol accumulates PI4P in intracellular dots. $n = 81$ cells from 27 roots (WT + DMSO, WT + E2), 75 cells from 25 roots (*IPCS1;2^amiRNA* + DMSO), 84 cells from 28 roots (*IPCS1;2^amiRNA* + E2). **c** Colocalization between PI4P biosensors mCIT-2×PH^FAPP1 or mCIT-3×PH^FAPP1 and the SVs/TGN marker VHA-a1-mRFP or ECHIDNA protein, respectively, in the *IPCS1;2^amiRNA* line induced by β-estradiol. **d** Colocalization between IPCS2-tagRFP and the SVs/TGN marker ECHIDNA, the CCVs/TGN marker CHC, or the Golgi marker MEMBRIN11. **e** Colocalization quantification of **c** and **d**. IPCS2 strongly co-localizes with SVs/TGN and more weakly with CCVs/TGN or the Golgi. PI4P biosensors strongly co-localize with SVs/TGN in the *IPCS1;2^amiRNA* line, to a similar level as the colocalization of IPCS2 with SVs/TGN. $n = 30$ roots (IPCS2 + ECH, IPCS2 + MEMB11, IPCS2 + CHC), 58 cells from 20 roots (2×PH^FAPP1 + VHA-a1), 60 cells from 20 roots (3×PH^FAPP1 + ECH). Statistics were done by two-sided Wilcoxson's rank-sum test (**b**) or two-sided Dwass-Steel-Critchlow-Flinger multiple comparison test with Monte Carlo method (10,000 iterations) (**e**), *$P < 0.01$, ***$P < 0.0001$. Each element of the boxplot indicates the following value: center line, median; box limits, the first and third quartiles; whiskers, 1.5× interquartile range; points above or below the whiskers, outliers. Scale bars, 10 μm.

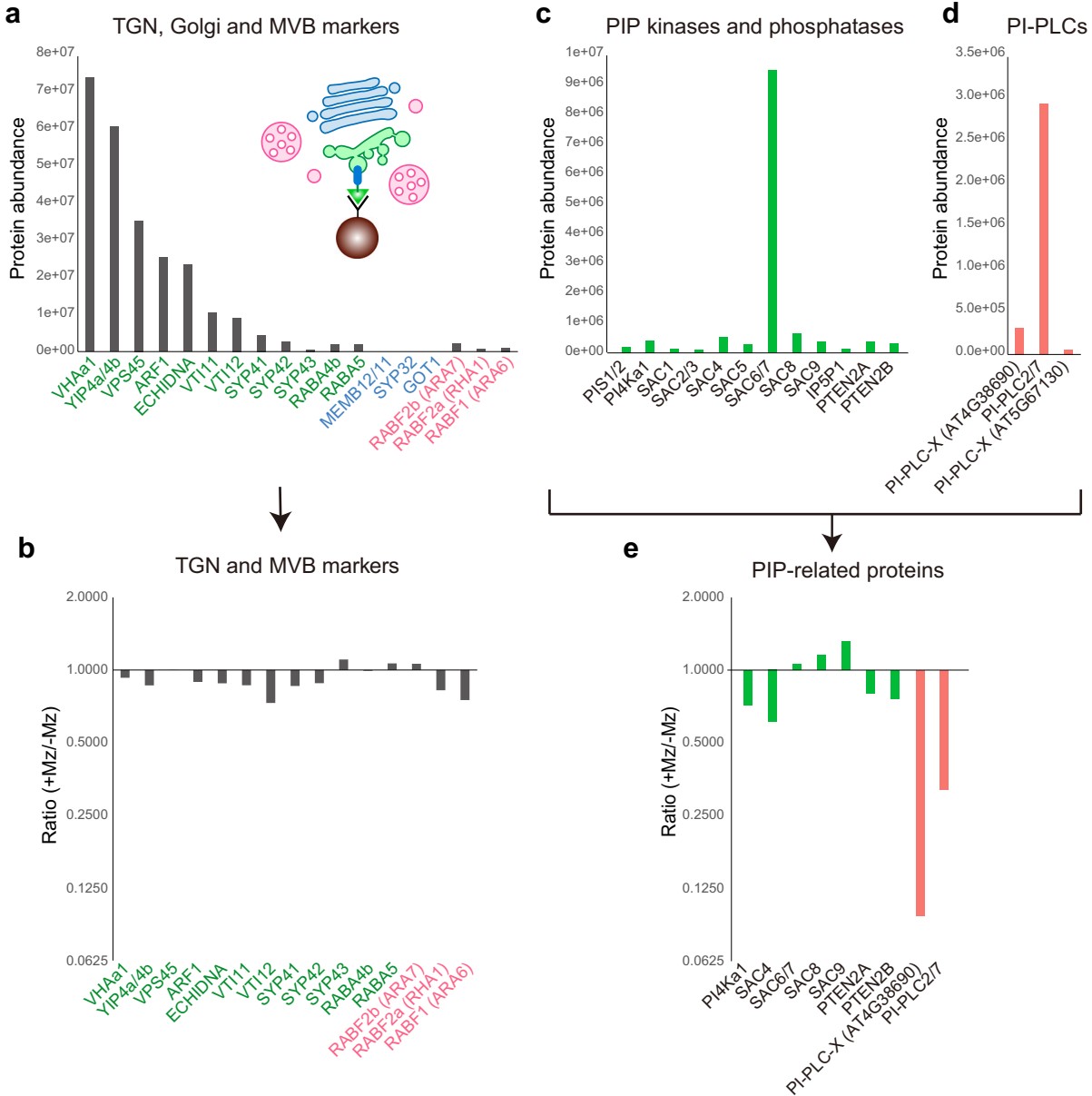

**Fig. 8 LC-MS/MS label-free proteomics identified PI-PLCs as potential target of VLCFAs at TGN. a, c, d, e** Protein abundance in SYP61-SVs/TGN immuno-purified compartments analyzed by label-free quantitative proteomics (detailed analysis is displayed in Supplementary Data 3 and synthesized in Supplementary Data 4, including the number of peptides found for each proteins). **a** TGN (green), Golgi (blue), and MVB (red) markers, **c** PIP kinases and phosphatases, **d** PI-PLCs. The abundance ratio of the proteins shown in **a** or **c, d** between control and Mz-treated samples is displayed in **b** or **e**, respectively. For the calculation of ratio, we applied a threshold of minimal protein amount as 300,000 to select the proteins from which we can get reliable ratio. TGN markers were abundant compared to Golgi and MVB markers, which indicates an efficient purification, and their abundance was not affected by metazachlor (Mz) treatment. In contrast, PI-PLCs were strongly reduced upon Mz treatment. $n = 4$ biological replicates.

higher amount of PI4P and PI(4,5)P$_2$, suggesting that PI4P and PI(4,5)P$_2$ are the in planta substrates of PI-PLCs[51]. PI-PLC is a family of nine members in *Arabidopsis*. Our proteomic results indicate that PI-PLC2 and PI-PLC7 might act in PI4P homeostasis at TGN. As PLC2 is a main player in the root, we chose to focus on PLC2. A previous membrane fractionation study reported that PLC2 is mostly localized at the PM and to a lesser extend in the microsomal fraction and small dots inside the cell[51,52]. To check whether PLC2 could localize at TGN, we produced a GFP-tagged version PLC2 driven by the pUB-IQUITIN10 promoter to get strong and uniform expression pattern. The signal was mostly localized at the PM but was also weakly present in intracellular dots (Supplementary Fig. 9a). Due

to the low signal intensity at dots, we processed the images with a Gaussian blur filter and subtracted the background. The processed images revealed intracellular dots partly colocalizing with the TGN marker VHA-a1-mRFP (Fig. 10e, Supplementary Fig. 9b). However, due to the low TGN signal in control condition, we could not confirm that PLC2 was deprived from TGN upon Mz treatment. In any case, a lower amount of PI-PLCs at SVs/TGN could result in a higher amount of PI4P at the TGN.

**SLs-mediated PI4P homeostasis involves PI4P consumption by PI-PLCs and impacts PIN2 sorting at SVs/TGN.** Next, we checked whether increased PI4P level could be due to less PI4P

**a** mCIT-2xPH$^{FAPP1}$

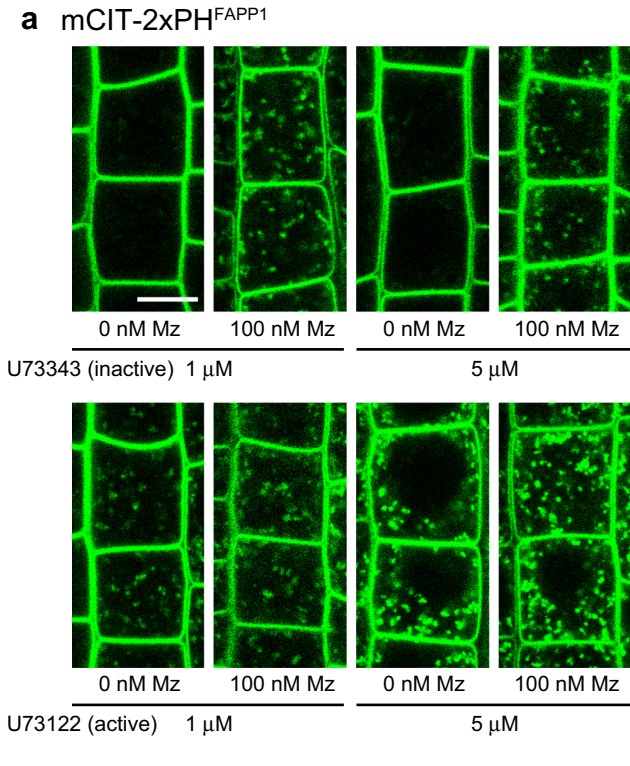

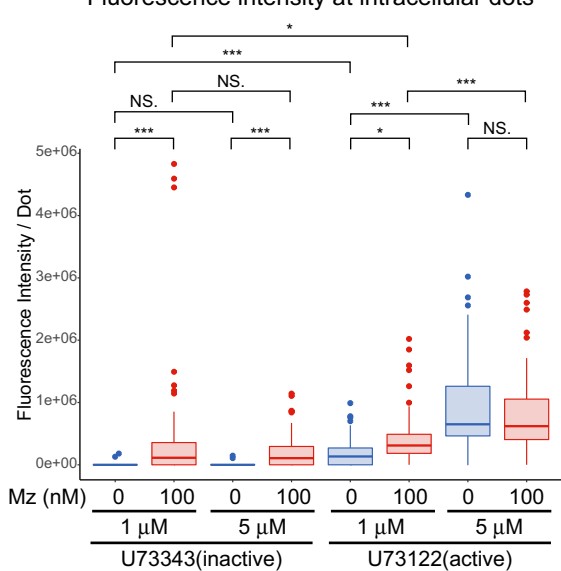

**b** Fluorescence intensity at intracellular dots

**Fig. 9 PI-PLC activity mediates the effect of acyl-chain length of sphingolipids on PI4P at the TGN. a** Confocal images of root epidermal cells expressing the mCIT-2×PH$^{FAPP1}$ PI4P biosensor treated with either the PI-PLC-inhibitor U73122 or its inactive analog U73343 at 1 or 5 μM on seedlings grown on 0 or 100 nM metazachlor (Mz)-containing medium. Scale bar, 10 μm. **b** Quantification of the fluorescence intensity at intracellular dots. The PI4P pool at dots was increased upon active U73122 treatment but not with inactive U73343, and this effect was enhanced by increasing the U73122 concentration from 1 to 5 μM. Treatment with 1 μM active U73122 on seedlings grown on 100 nM Mz increased PI4P pool at intracellular dots but not with 5 μM active U73122 indicating that Mz effect is mediated by PI-PLCs. $n = 66$ cells from 22 roots (0 nM Mz + 1 μM U73343, 100 nM Mz + 5 μM U73343, 100 nM Mz + 1 μM U73122, 0 nM Mz + 5 μM U73122), 63 cells from 21 roots (100 nM Mz + 1 μM U73343, 0 nM Mz + 1 μM U73122, 100 nM Mz + 5 μM U73122), 60 cells from 20 roots (0 nM Mz + 5 μM U73343). Statistics were done by two-sided Dwass-Steel-Critchlow-Flinger multiple comparison test with Monte Carlo method (10,000 iterations). *$P < 0.01$, *** $P < 0.0001$. Each element of the boxplot indicates the following value: center line, median; box limits, the first and third quartiles; whiskers, 1.5× interquartile range; points above or below the whiskers, outliers.

consumption as our proteomic analysis identified a potential implication of PI-PLCs in Mz-induced accumulation of PI4P at the TGN (Fig. 8d, e). We used the mCIT-2×PH$^{FAPP1}$ PI4P biosensor upon 90 min treatment with either the PI-PLC-inhibitor U73122 or its inactive analog U73343 at 1 μM and 5 μM for both active and inactive analogs[53,54] (Fig. 9a). We did not observe any significant change in fluorescence intensity at intracellular dots between 1 μM and 5 μM of inactive U73343 control treatment (Fig. 9b). Contrastingly, the active U73122 PI-PLCs inhibitor treatment clearly displayed a significant increase of PI4P sensor at 1 μM and this effect was further increased at 5 μM (Fig. 9b). These results suggest that PI-PLCs play a role on PI4P homeostasis in intracellular compartments. Furthermore, when seedlings were grown on 100 nM Mz prior to treatment with 1 μM of

active U73122, the fluorescence intensity of PI4P sensor in intracellular compartments got even stronger (Fig. 9b). However, with 5 μM of active U73122 PI-PLCs inhibitor, there was no significant difference between seedlings grown on either 0 or 100 nM Mz (Fig. 9b). These results argue that SLs act in PI4P accumulation at the TGN through the modulation of a pool of PI-PLC proteins. To unambiguously identify the nature of the intracellular compartments in which PI4P sensors accumulates, we performed colocalization of the mCIT-2×PH$^{FAPP1}$ PI4P biosensor with the SVs/TGN markers VHA-a1 fused to mRFP and ECHIDNA[55,56]. Our results revealed a strong colocalization between mCIT-2× or mCIT-3×PH$^{FAPP1}$ and either VHA-a1-mRFP or ECHIDNA upon Mz treatment at a similar level to what we observed in *IPCS1;2amiRNA* background, confirming that the intracellular accumulation of PI4P sensors occurs at SVs/TGN when the acyl-chain length of SLs is reduced (Fig. 10a, e). Although a little weaker, we identified a similar level of colocalization between mCIT-2× or mCIT-3×PH$^{FAPP1}$ and either VHA-a1-mRFP or ECHIDNA upon inhibition of PI-PLC by U73122 treatment as well (Fig. 10a, e). Next, we confirmed our PLC-inhibitor findings using a knockout mutant allele of *PI-PLC2*[57]. Although the mutant has gametophytic lethality and produces drastically less homozygotes plants (that do not produce any seeds) than the wild-type, we could still get a segregating F2 *pi-plc2* mutant population crossed with the 2×PH$^{FAPP1}$/PI4P biosensor. Our results revealed that the levels of PI4P biosensor is increased both at intracellular dots and the PM in *pi-plc2* mutant (Supplementary Fig. 9c, d). These findings are consistent with the function and localization of PLC2 at both the PM and the TGN. Altogether, our pharmacological, genetic, and localization results consistently support a role of PI-PLCs in consuming PI4P at the TGN.

Both PI-PLCs and acyl-chain length of sphingolipids are known to be involved in root gravitropism, auxin distribution, and PIN2 apical polarity[22,53,57]. However, whether PI-PLCs are involved in PIN2 sorting at SVs/TGN is not known. Hence, we thoroughly quantified the fluorescence intensity of PIN2, specifically at intracellular dots, upon inhibition of PI-PLC by active U73122 treatment or upon treatment with its inactive analog U73343. Our results revealed no difference between the control condition and treatment with the inactive PI-PLC-

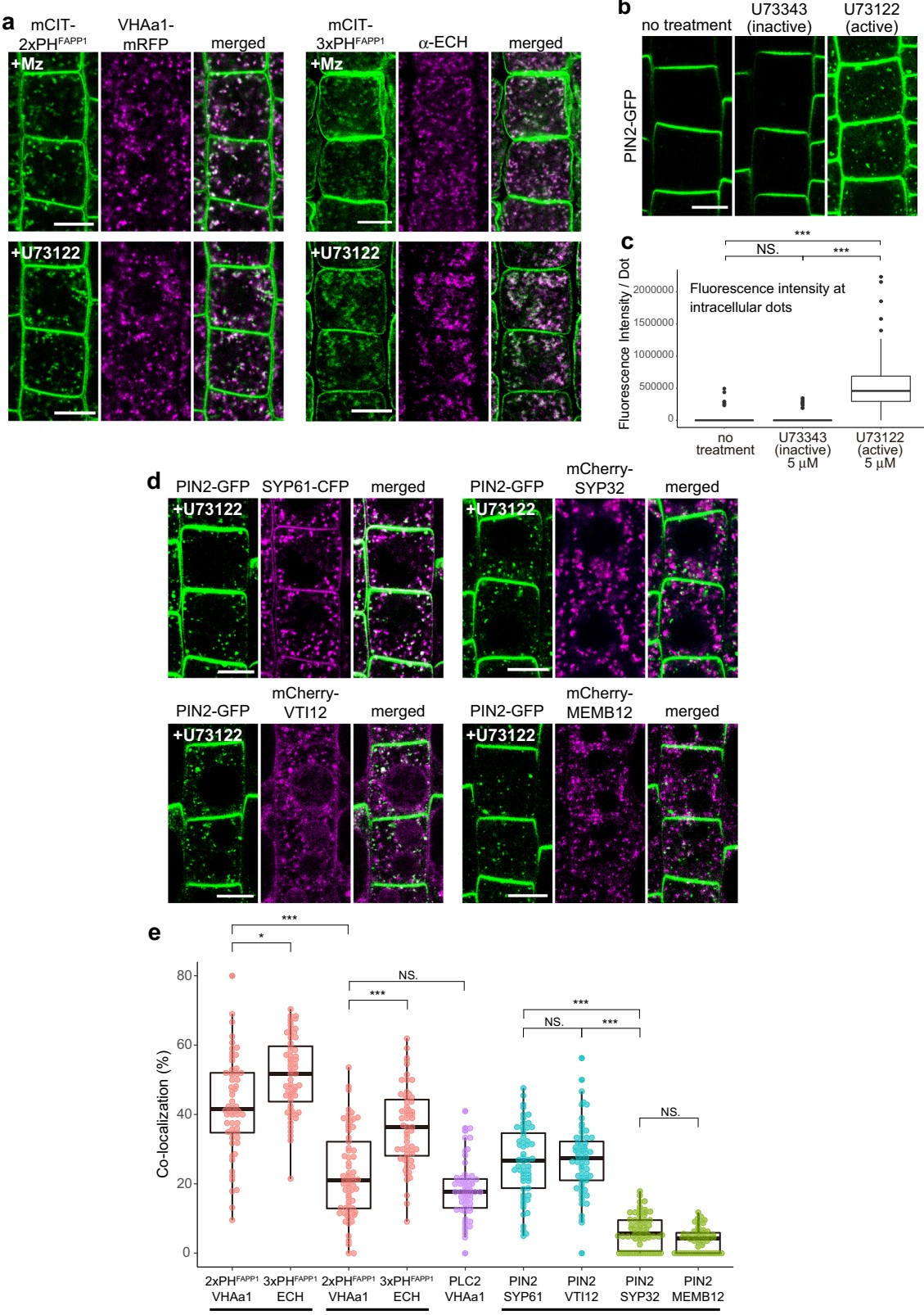

inhibitor analog U73343 (Fig. 10b, c). However, a strong accumulation of PIN2 in intracellular compartments was observed upon treatment with 5 μM of active PI-PLC-inhibitor U73122 (Fig. 10b, c). To identify the compartments in which PIN2 accumulates, we performed colocalization experiments between PIN2 and the SVs/TGN-localized SYP61-CFP syntaxin or VTI12-mCherry SNARE markers. The level of colocalization we quantified between PIN2 and SVs/TGN markers was similar to what we quantified between PI4P sensors and the SVs/TGN markers VHA-a1-mRFP or ECHIDNA upon the active PI-PLC-

**Fig. 10 SL-mediated modulation of PI4P through PI-PLC impacts PIN2 sorting at SVs/TGN. a** Colocalization between PI4P biosensors mCIT-2×PH[FAPP1] or mCIT-3×PH[FAPP1] and the SVs/TGN marker VHA-a1-mRFP or ECHIDNA protein, respectively. Seedlings were either grown on 100 nM metazachlor (Mz) (upper panels) or treated with 5 µM PI-PLC-inhibitor U73122 (lower panels). In both cases (Mz or U73122 treatment), the PI4P-positive intracellular dots showed strong colocalization with TGN. **b** Confocal images of root epidermal cells expressing PIN2-GFP upon control condition (left), with 5 µM inactive analog of PI-PLC inhibitor (U73343, middle), and 5 µM active PI-PLC inhibitor (U73122, right). **c** Quantification of the fluorescence intensity of PIN2-GFP specifically at intracellular dots showed that PIN2 was accumulated at endomembrane compartments upon U73122 treatment. $n = 69$ cells from 23 roots (no treatment), 63 cells from 21 roots (U73343), 66 cells from 22 roots (U73122). **d** Colocalization of endomembrane compartments labeled by PIN2-GFP upon 5 µM U73122 treatment with either SVs/TGN labeled by SYP61-CFP (upper left), SVs/TGN labeled by mCherry-VTI12 (lower left), Golgi apparatus labeled by mCherry-SYP32 (upper right), or Golgi apparatus labeled by mCherry-MEMB12 (lower left). **e** Colocalization quantification of **a** and **d**. PIN2 showed significantly higher colocalization with SVs/TGN markers compared to the Golgi markers. $n = 60$ cells from 20 roots (2×PH[FAPP1] × VHA-a1 + Mz, 3×PH[FAPP1] × ECH + Mz, PIN2 × VTI12 + U73122), 59 cells from 20 roots (3×PH[FAPP1] × ECH + U73122), 69 cells from 24 roots (2×PH[FAPP1] × VHA-a1 + U73122), 60 cells from 21 roots (PLC2 × VHA-a1), 57 cells from 22 roots (PIN2 × SYP61 + U73122), 66 cells from 23 roots (PIN2 × SYP32 + U73122), 55 cells from 21 roots (PIN2 × MEMB12 + U73122). Statistics were done by two-sided Dwass-Steel-Critchlow-Flinger multiple comparison test with Monte Carlo method (10000 iterations), $*P < 0.01$, $***P < 0.0001$. Each element of the boxplot indicates the following value: center line, median; box limits, the first and third quartiles; whiskers, 1.5× interquartile range; points above or below the whiskers, outliers. Scale bars, 10 µm.

inhibitor U73122 (Fig. 10d, e). Importantly, we quantified a weak colocalization between PIN2 and the Golgi-localized SYP32 syntaxin and MEMBRIN12 SNARE markers upon U73122 (Fig. 10d, e). Taken together our results show that impacting either the acyl-chain length of sphingolipids or PI-PLC function lead to similar phenotypes including (1) an accumulation of PI4P at the TGN, (2) mis-sorting of PIN2 at SVs/TGN, and (3) root gravitropic defects. Because Mz reduces the amount of PI-PLC at the TGN, it is likely that the direct consumption of PI4P by PI-PLCs within this compartment explains the cellular and plant phenotypes induced by this treatment.

## Discussion

The homeostasis of phosphoinositides at the TGN is crucial for the regulation of membrane trafficking and the sorting of cargos. The interplay between sphingolipids and phosphoinositides has been evidenced in mammals at ER/TGN contact sites and involves a complex regulatory homeostatic loop based on OSBP-mediated PI4P/sterols or PI4P/PS lipid exchange mechanisms induced by the grafting of a polar head on ceramide at the TGN[5,6,10]. In this study, we identified that the acyl-chain length of sphingolipids is a crucial factor regulating the homeostasis of PI4P at the TGN of plant cells. Our results show that sphingolipids do not impact PI4P through TGN-localized PI4-kinases. We could further characterize that this effect is independent from PS at the TGN and does not impact sterol quantity at the TGN. Although we cannot completely exclude the possibility that the effect of sphingolipids on PI4P involves lipid exchange mechanisms induced by sphingolipid-metabolic flux, our results strongly suggest that the interplay between sphingolipids and phosphoinositides originates from another mechanism. Indeed, our work provides a mechanistic connection between sphingolipids and PI4P at the TGN through sphingolipid-mediated consumption of PI4P by PI-PLC. PI-PLCs are well recognized to catalyze hydrolysis of PI(4,5)P₂, but *Arabidopsis* PLCs hydrolyze equally well both PI4P and PI(4,5)P₂ in vitro, and PI4P is much more abundant in vivo compared to PI(4,5)P₂[27,58,59]. Moreover, the involvement of PI-PLCs in modulating the global quantity of PI4P has also been previously reported in planta[51]. Our results have now uncovered an unexpected role of sphingolipid-mediated PLCs in sorting of the auxin efflux carrier PIN2 at SVs of TGN. Consistently with our findings, both acyl-chain length of sphingolipids and PI-PLCs are known to be involved in PIN2 polarity, auxin distribution and root gravitropism[22,53,57,60]. Our results suggest that VLCFAs of glycosylated sphingolipids (GIPCs and/or GluCer) are involved in PI4P homeostasis and PIN2 sorting at TGN rather than VLCFAs of ceramides. Interestingly, the sorting of the auxin efflux carrier PIN1, but not

PIN2, has been shown to be dependent on VLCFAs of ceramides at RAB-A2a-positive/CCVs compartments[38]. Thus, it is reasonable to assume that different species of sphingolipids would be differentially distributed between SVs and CCVs of TGN and be involved in distinct sorting mechanisms[37]. The localization of IPCS enzymes at SVs would be a way to segregate these mechanisms. In the future, addressing these issues would greatly enhance our understanding of how TGN is able to differentiate trafficking pathways and polar sorting. Additionally, understanding how sphingolipids act on PI-PLCs would be of great interest to the whole community. Although the involvement of PI-PLC in protein sorting from the TGN to the PM has been proposed in animal cells[61,62], their integration in lipid interplay has never been hypothesized while regulation of PI4P quantity at the TGN is instrumental in the face of extensive membrane trafficking and fluctuating lipid metabolic fluxes. In animal cells, a few studies suggested the existence of a crosstalk between sphingolipids and PI-PLC, such as the hypersensibility of the GPI-anchored protein CD14 to PI-PLC upon sphingolipids-deficiency in mammals[63] or more recently the role of ceramide kinase CERK in regulating PI-PLC signalling and the formation of PI(4,5)P₂ clustering into membrane ordered domains during phototransduction in *Drosophila*[64]. Our work identified a function of sphingolipids in PI4P consumption at the TGN and subsequently in the polar sorting of PIN2. Whether this mechanism is common to eukaryotic organisms represents future perspectives in understanding the function of lipid interplay during protein sorting and cell polarity.

## Methods

**Plant material and growth conditions.** The following *Arabidopsis* transgenic fluorescent protein marker lines were used: PI4P biosensors pUBQ10::mCITRINE-1×PH[FAPP1] (P5Y)[27], pUBQ10::mCITRINE-2×PH[FAPP1] (P21Y)[27], pUBQ10::mCI-TRINE-3×PH[FAPP1] [27], pUBQ10::mCITRINE-1×PH[FAPP1]-E50A[24], pUBQ10::mCI-TRINE-1×PH[FAPP1]-E50A-H54A[24], and pUBQ10::mCITRINE-P4M[SidM] [24]. PS biosensor pUBQ10::mCITRINE-C2[LACT] [24]. TGN markers pARF1::ARF1-GFP[30], pSYP61::SYP61-CFP[65], pVHA-a1::VHA-a1-mRFP[55], and pUBQ10::mCherry-VTI12 (W13R)[66]. Golgi markers pUB10::mCherry-SYP32 (W22R)[66] and mCherry-MEMB12 (W127R)[66]. Auxin efflux carrier PIN2 marker pPIN2::PIN2-GFP[30]. Additionally, *kcs9*, *pi4kβ1;pi4kβ2* double mutant[28], *pi-plc2* (SALK_152284)[60], *ipcs1; ipcs2* double mutant [crossing of *ipcs1* (SALK_087676) and *ipcs2* (FLAG_156E02), kept as hemizygote for *ipcs2*], and *IPCS1;2* artificial microRNA line (described below) were used for the analyses in KO or KD background. pUBQ10::PLC2-GFP and pIPCS2::IPCS2-TagRFP were directly transformed into pVHA-a1::VHA-a1-mRFP fluorescent line and *ipcs1;ipcs2* double mutant line, respectively. The other double fluorescent lines and mutant background fluorescent lines were established by crossing of the lines above, except for mCITRINE-2×PH[FAPP1] in *pss1-3* that has been published beforehand[25]. For confocal observations without drug treatment, sterilized seeds were kept at 4 °C in water for 2–3 days, sown on half Murashige and Skoog (MS) agar medium plates (0.8% plant agar, 1% sucrose, and 2.5 mM mor-pholinoethanesulfonic acid pH5.8 with KOH), and grown in 16 h light/8 h

darkness at 22 °C for 5 days. The growth condition for membrane compartment immunoprecipitation is described hereafter.

**Cloning and plant transformation**. For the co-expression of PLC2-GFP with VHA-a1-mRFP, the binary vector pUBQ10::PLC2-eGFP was constructed by using the multisite gateway system. *Arabidopsis PLC2* (At3g08510) genomic sequence was amplified by PCR and introduced into pDONR221 (Thermo Fisher Scientific) by BP recombination. Similarly, the DNA sequence encoding eGFP, which is optimized for *Arabidopsis* codon usage was introduced into pDONR P2RP3 (Thermo Fisher Scientific). These plasmids and UBIQUITIN10 promoter in pDONR P4P1R[67] were used for LR recombination into a gateway destination vector pB7m34GW[68].

For the expression of IPCS2-TagRFP driven by the native promoter in *ipcs1; ipcs2* double mutant, the genomic sequence encoding *IPCS2* (At2g37940) was amplified by PCR with additional AatII restriction enzyme cutting site at 5' end and integrated into pENTR/D-TOPO vector (Thermo Fisher Scientific). For the promoter, the 2000 bp of the 5' region of *IPCS2* gene was amplified by PCR with additional AatII sites at both 5' and 3' ends, and integrated into pGEM-T Easy vector (Promega) by TA cloning system. The promoter fragment was taken by AatII digestion from pGEM-T Easy vector and ligated into pENTR/D-TOPO vector containing *IPCS2* gene. The resulted *pIPCS2::IPCS2* fragment was recombined by LR reaction into pGWB559[69], which contains TagRFP gene at the 3' side of the recombination site.

For the inducible *IPCS1;2* artificial microRNA line, the miRNA sequence targeting both *IPCS1* (At3g54020) and *IPCS2* (At2g37940) was designed on WMD2 (WebMicroRNA Designer2). Using pRS300 vector[70] as template, we exchanged the natural *Arabidopsis* miR319a sequence with the designed amiRNA by overlapping PCR using I miR-s-1, II miR-a-1, III miR*s-1, IV miR*a-1, A and B primers. The resulted amiRNA was introduced into pENTR/D-TOPO (Thermo Fisher Scientific), and further recombined by LR reaction into pMDC7[71], which contains the β-estradiol inducible promoter.

These binary plasmids were transformed into *Agrobacterium tumefaciens* and used for the transformation into *Arabidopsis* plants by floral dipping. The transformants were selected by Basta (*pUBQ10::PLC2-GFP*) or hygromycin (*pIPCS2::IPCS2-TagRFP* and *IPCS1;2amiRNA*).

All primers used for cloning are described in Supplementary Data 2.

**Gravitropism assay**. Gravitropism assay was performed as described in[22]. Briefly, *Arabidopsis* seedlings were grown on half MS agar plates vertically as described in "Plant materials and growth conditions" section for 3 days, and then transferred to darkness under the same growth conditions and incubated during 24 h, maintaining the same growth plate orientation. Next, the plates were turned 90° counter-clockwise and incubated vertically in the dark for 24 h. The plates were then photographed, and the last curvature of the root was measured using imageJ. The new gravity vector was labelled as 0 while all the root angles were ranked into 24 classes of 15° angles for quantification.

**Inhibitor treatments**. Mz treatment for confocal observations except for the fatty acid add-back experiment was performed on seedlings grown for 5 days on half MS plates containing 10, 50, or 100 nM Mz (Cayman Chemical). Mz was added to the medium from a 100 mM stock in dimethylsulfoxide by using an intermediate diluted stock at 100 µM (extemporarily prepared). The treatments for the fatty acid add-back and SVs/TGN immuno-purification experiments are described hereafter. FB1 treatment for the confocal observations was performed by transferring the seedlings grown on drug-free half MS plates into the liquid half MS medium containing 2.5 µM FB1 (Sigma–Aldrich) 20 h before observation. For the sphingolipid analysis, plants were grown on half MS agar plates containing 0.5 µM FB1 for 5 days. FB1 stock solution was prepared in dimethylsulfoxide at 0.5 mM. β-Estradiol treatment for amiRNA induction was performed by growing the seedlings on half MS agar plates containing 5 µM β-Estradiol (Sigma–Aldrich) for 5 days. PI-PLC-inhibitor treatment (U73122 and its inactive analog U73343, Sigma–Aldrich) was performed on seedlings grown on drug-free half MS plates for 5 days and transferred in liquid half MS medium containing 1 or 5 µM of either U73122 or U73343 for 90 min. The drug concentrations are also indicated in the figure legends.

**Fatty acid add-back**. For the treatment, media containing fatty acids were prepared by adding h24:0 (2-Hydroxytetracosanoic acid, Matreya) or 24:0 (Tetracosanoic acid, Matreya) at 50 µM to liquid half MS medium and heating at 70 °C for 30 min. The fatty acids were added from 25 mM stock solutions in chloroform: methanol (5:1) solvent mix. After heating, the media were cooled to the room temperature and 50 nM Mz was added. *Arabidopsis* seedlings were grown on half MS plates without Mz and transferred to the liquid media containing fatty acid and Mz. They were incubated for 48 h with mild shaking under 16 h light/8 h darkness at 22 °C. The plants were directly used for the confocal imaging after treatment. For the fatty acid quantification, the seedlings were washed by 30 mL of half MS for three times after treatment in order to remove the remaining fatty acid on the plant surface, and only the roots were collected. The fatty acid analysis is described hereafter.

**Immunocytochemistry**. Whole-mount immunolabelling of *Arabidopsis* root was performed as described[22]. In brief, seedlings were fixed in 4% paraformaldehyde dissolved in MTSB (50 mM PIPES,5 mM EGTA, 5 mM MgSO₄ pH 7 with KOH) for 1 h at room temperature (RT) and washed three times with MTSB. Roots were cut on SuperFrost Plus Gold slides (Menzel-Gläser) and dried at RT. Roots were then permeabilized with 2% Driselase (Sigma) dissolved in MTSB for 30 min at RT, rinsed four times with MTSB, and treated with 10% dimethylsulfoxide + 3% Igepal CA-630 (Sigma) dissolved in MTSB for 1 h at RT. Nonspecific sites were blocked with 5% normal donkey serum (NDS, Sigma) in MTSB for 1 h at RT. Primary antibodies, in 5% NDS/MTSB, were incubated overnight at 4 °C and then washed four times with MTSB. Secondary antibodies, in 5% NDS/MTSB, were incubated 1 h at RT and then washed four times with MTSB. Antibody dilutions were as follows: rabbit anti-ECH[56] 1/300, rabbit anti-CHC (Agrisera) 1/300, rabbit anti-MEMB11[72] 1/300, TRITC-coupled donkey anti-rabbit IgG (Jackson Immunoresearch) 1/300 and AlexaFluor 647 (A647)-coupled donkey anti-rabbit IgG (Jackson Immunoresearch) 1/300.

**Confocal microscopy and image analyses**. Confocal laser scanning microscopy was performed using a Zeiss LSM 880. Seedlings were mounted with 1/2 MS medium (with or without drugs). Double-sided tape was used as the spacer to separate the slide glass and coverslip. All acquisitions were done with an oil-immersion ×40 objective, 1.3 numerical aperture (APO ×40/1.3 Oil DIC UV-IR). For the observations of mCITRINE-2×PH^FAPP1 in *pss1-3* or *plc2* mutant backgrounds, the homozygote mutant plants were selected by their shorter and agravitropic root phenotypes.

Quantification of the fluorescence intensity was performed by ImageJ. For the PM, the outline of the cell was drawn outside of the PM by hand, and the signal intensity was quantified in the region that is within 1.5 µm inside the outline, which was subsequently normalized by the area. For the intracellular dots, mask images were created by subtracting background (rolling ball radius = 10 pixels, ~1 µm) and applying threshold, and those masks were used to extract the dots in the cytoplasmic area from the original images. The total signal intensity was normalized by the number of the dots. The threshold was kept constant for all the samples that are shown in the same graph. In order to avoid including nondotty background, only the structures with the circularity over 0.1 were quantified (circularity is defined as $4\pi A/P^2$ with $A$ = area and $P$ = perimeter, and it takes the values 0.0–1.0 with 1.0 represents the perfect circle). For the experiments using PI-PLC inhibitor (Figs. 9b and 10c, e), a size filter of 10–400 pixels (~0.1–4 µm²) was additionally applied.

Colocalization analyses were performed using the geometrical (centroid) object-based method[22]. The cells were segmented by hand and subcellular compartments were extracted by applying subtract background (rolling ball radius = 10 pixels, ~1 µm) and a threshold, and the distance between the centroids of two objects was calculated using 3D objects counter plugin of imageJ. For the analysis of pUBQ10::PLC2-GFP in VHA-a1-mRFP, Gaussian blur filter was applied before subtracting background in order to remove the homogenous signal in the cytosol. When the distance between two labelled structures was below the optical resolution limit, the colocalization was considered as true. The resolution limit was calculated based on the shorter emission maximum wavelength of the fluorophores. For the better extraction of the subcellular compartments, a size filter of 10–400 pixels (~0.1–4 µm²) was applied.

**Expression analysis by RT-qPCR**. For the expression analysis of *IPCS1* and *IPCS2* in the inducible amiRNA line, total RNA was extracted from roots 5 days after germination using the RNeasy Plant Mini kit (Qiagen) according to the manufacturer's instructions. Roots were disrupted using stainless steel beads 5 mm (Qiagen) and Tissuelyser II (Qiagen). First strand cDNA was synthesized on 2 µg of total RNA by SuperScript II Reverse Transcriptase (Thermo Fisher Scientific) using oligo(dT) in a final volume of 20 µL. Then, mRNA was treated with DNase I using DNa-free Kit (Thermo Fisher Scientific). 4 ng of cDNA and SYBR Green Master Mix (Roche) was employed for a LightCycler 480 Real-Time PCR System from Roche. The transcript abundance in samples was determined using a comparative cycle threshold (Ct) method. The relative abundance of the reference genes PEX4 (AT5G25760) and At4g33380 mRNAs in each sample was determined and used to normalize for differences of total RNA level. All primers used for RT-qPCR are described in Supplementary Data 2. Data were analyzed using the GEASE software developed at the Magendie Neurocenter.

**Immunoprecipitation of intact SVs/TGN**. Immunoprecipitation of intact SVs/TGN membrane compartments was performed as described[22,73]. In brief, *Arabidopsis* seedlings were grown in flasks with liquid half MS medium for 9 days under 120 rpm, shaking and 16 h light/8 h darkness cycle. For the Mz-treated samples, 100 nM Mz was added to the culture at the day 4. Seedlings were grinded by an ice-cooled mortar and pestle in vesicle isolation buffer (HEPES 50 mM pH 7.5, 0.45 M sucrose, 5 mM MgCl₂, 1 mM dithiothreitol, 0.5% polyvinylpyrrolidone, and 1 mM phenylmethylsulfonyl fluoride). The homogenate was filtered and centrifuged to remove the debris. The supernatant was loaded on 38% sucrose cushion (dissolved in 50 mM HEPES pH 7.5) and centrifuged at 150,000 × $g$ for 3 h at 4 °C. After removing the supernatant above the membrane pool located at the interface

between the sucrose and the supernatant, a step-gradient sucrose was built on the top of the membrane interface with successive 33% and 8% sucrose solutions (dissolved in 50 mM HEPES pH 7.5). The tubes were centrifuged overnight at 150,000 × g at 4 °C, the membranes that appeared at the 38/33% and 33/8% sucrose interfaces were harvested, pooled and diluted in 2–3 volume of 50 mM HEPES pH 7.5. After a centrifugation step at 150,000 × g for 3 h at 4 °C, the pellet was resuspended in resuspension buffer (50 mM HEPES pH 7.5, 0.25 M sucrose, 1.5 mM MgCl$_2$, 150 mM NaCl, 1 mM phenylmethylsulfonyl fluoride, and protease inhibitor cocktail from Sigma–Aldrich). The protein amount of the resuspended membrane fractions was quantified by Bicinchoninic Acid Protein Assay Kit (Sigma–Aldrich) and equilibrated between different samples. Those equilibrated membrane fractions were used as the IP input. IP was performed with magnetic Dynabeads coupled to proteinA (Thermo Fisher Scientific) conjugated with rabbit anti-GFP antibody (Thermo Fisher Scientific) by bis[sulfosccinimidyl] suberate (Thermo Fisher Scientific). The beads were incubated with the IP input for 1 h at 4 °C, washed and resuspended in resuspension buffer.

**Western blotting of IP fractions.** To equally load the IP input and the beads-IP fraction (IP output) for Western blotting, TGX Stain-Free FastCast premixed acrylamide solution (Bio-Rad) and ChemiDoc MP imaging system (Bio-Rad) were used to visualize the proteins. The whole individual lanes were quantified by ImageJ software and the quantity of proteins were adjusted to get equal new loading between each lane. Western blotting was performed with mouse anti-GFP (1/1,000, Roche, 118144600001) and rabbit anti-ECH[56] (1/1,000) as the primary antibodies, and goat anti-mouse IgG-HRP conjugate (1/3,000, Bio-Rad, 1721011) and anti-rabbit IgG-HRP conjugate (1/5,000, Bio-Rad, 1706515) as the secondary antibodies.

**Characterization of lipid composition (fatty acids, sphingolipids, and phosphoinositol monophosphate).** Fatty acids and sterols characterization of immuno-precipitated intact TGN compartments is described in[22,73]. Shortly, for the characterization of fatty acids, 150 μl of IP beads fraction was incubated with 1 ml of 5% sulfuric acid in methanol, including 5 μg/ml of the lipid standards C17:0 to normalize non-hydroxylated fatty acids and h14:0 to normalize α-hydroxylated fatty acids. The transesterification was performed overnight at 85 °C. After cooling down at room temperature, the fatty acids methyl esters (FAMEs) were extracted by adding 1 ml of NaCl 2.5% and 1 ml of hexane. After hand shaking and centrifugation at 700 × g for 5 min at room temperature, the higher phase was collected and placed in a new tube. After addition of 1 ml of 100 mM Tris, 0.09% NaCl pH 8 with HCl, hand shaking and centrifugation, the higher phase was collected and evaporated. After evaporation, 200 μl of N,O-Bis(trimethylsilyl)trifluoroacetamide +1% trimethylsilyl (BSTFA + 1% TMCS, Sigma) were added and incubated at 110 °C for 15 min. After evaporation, lipids were resuspended in 80 μl of 99% hexane. For the characterization of sterols, 50 μl of IP beads fraction was incubated with 1 ml of chloroform:methanol 2:1, including 5 μg/ml of the sterol standard α-cholestanol, for 2 h at room temperature. After addition of 0.9% (w/v) NaCl, hand shaking and centrifugation, the lower organic phase was collected and evaporated. Fatty acids were removed by saponification by adding 1 ml of 99% ethanol and 100 μl of 11 N KOH for 1 h at 80 °C. Then, 1 ml of 99% hexane and 2 ml of water were added. After hand shaking and centrifugation, the higher phase was collected, placed in a new tube where 1 ml of 100 mM Tris, 0.09% NaCl pH 8 with HCl. After hand shaking and centrifugation, the higher phase was collected, placed in a new tube and evaporated. After evaporation, 200 μl of BSTFA + 1% TMCS were added and incubated at 110 °C for 15 min. After evaporation, lipids were resuspended in 80 μl of 99% hexane. GC-MS was performed using an Agilent 7890 A and MSD 5975 Agilent EI with the following settings: the helium carrier gas was set at 2 ml/min, the splitless mode was used for injection, the temperatures of injector and auxiliary detector were set at 250 and 352 °C, respectively, the oven temperature was held at 50 °C for 1 min, a 25 °C/min ramp (2-min hold) and a 10 °C/min ramp (6-min hold) were programmed at 150 °C and 320 °C, respectively, the MS analyzer was set in scan only with a mass range of 40–700 m/z in positive mode with electron emission set to 70 eV, the MS source and the MS Quad were set to 230 and 50 °C, respectively. Compounds were identified against the NIST14 database and the area of the GC peaks were determined using MassHunter qualitative analysis software (Agilent).

Quantification of phosphatidylinositol monophosphate (PIPs) was performed accordingly to[74]. Briefly, 150 μl of IP beads fraction was incubated with 725 μl of methanol:chloroform:HCl 1 M 2:1:0.1 and 15 μl H$_2$O in presence of 10 ng of 17:0–20:4 PI4P standard (Avanti). Seven hundred and fifty microliters of chloroform and 170 μl HCl 2 M were then added. The samples were vigorously shaken and centrifuged. The lower phase was washed with 700 μl of the upper phase of a mix of methanol:chloroform:HCl 0.01 M 1:2:0.75, vortexed and centrifuged. The lipids of the lower phase were then methylated by adding 50 μl of TMS diazomethane for 10 min, the reaction is stopped by adding 6 μl of methanol: acetic acid 1:0.25. The samples were washed twice with 700 μl of the upper phase of a mix of methanol:chloroform:water 1:2:0.75, centrifuged and the lower phase collected. 100 μl of methanol:water 9:1 were added and the samples were concentrated under nitrogen flux up to get only a drop (10 μl) remaining. Finally, 80 μl of methanol were added to the samples, sonicated, and then 20 μL water were added to the samples and sonicated. LC-MS/MS (multiple-reaction-monitoring

mode) analyses were performed with a mass spectrometer QTRAP 6500 (ABSciex) mass spectrometer coupled to a liquid chromatography system (1290 Infinity II, Agilent). Analyses were achieved in positive mode; Nitrogen was used for the curtain gas (set to 30), gas 1 (set to 30), and gas 2 (set to 30). Needle voltage was at +5500 V with needle heating at 200 °C; the declustering potential was +100 V. The collision gas was also nitrogen; collision energy was set at +35 eV. The dwell time was set to 9 ms. Reverse-phase separations were carried out on a Jupiter C4 column (50 × 1 mm; particle size, 5 μm; Phenomenex). Eluent A was H$_2$O and 0.1% formic acid, and eluent B was acetonitrile and 0.1% formic acid. The gradient elution program was as follows: 0–2 min, 45% eluent B; 27 min, 100% eluent B; and 27–30 min, eluent 100% B. The flow rate was 100 μl/min; 10-μl sample volumes were injected. The areas of LC peaks were determined using MultiQuant software (ABSciex) for relative quantification to the area of the internal standard, values are expressed in arbitrary unit (AU).

Quantification of sphingolipids was performed as described in[75,76]. Briefly, lipids were extracted in 3 ml of propan-2-ol:hexane:water 2.75:1:1.25, incubated at 60 °C for 20 min, centrifuged and the supernatant was collected. This extraction procedure was repeated twice and the supernatants pooled in the same tube. After evaporation, the samples were treated with 2 ml of methylamine solution (7 ml of methylamine 33% in ethanol + 3 ml of methylamine 40% in water) to degrade all lipids that do not have an amide bound so that only sphingolipids remain. The methylamine treatment was performed at 50 °C for 1 h. After evaporation, the samples were resuspended in 90 μl of THF:methanol:water 2:1:2 with 0.1% formic acid containing synthetic internal lipid standards (Cer d18:1/C17:0, GluCer d18:1/C12:0 and GM1) was added, thoroughly vortexed, incubated at 60 °C for 20 min, sonicated 2 min and transferred into LC vials. LC-MS/MS (multiple-reaction monitoring mode) analyses were performed with a mass spectrometer QTRAP 6500 (ABSciex) mass spectrometer coupled to a liquid chromatography system (1290 Infinity II, Agilent). Analyses were performed in the positive mode. Nitrogen was used for the curtain gas (set to 20), gas 1 (set to 20), and gas 2 (set to 30). Needle voltage was at +5500 V with needle heating at 400 °C; the declustering potential was adjusted between +10 and +40 V. The collision gas was also nitrogen; collision energy varied from +15 to +60 eV on a compound-dependent basis. Reverse-phase separations were performed at 40 °C on a Supercolsil ABZ + , 100 × 2.1 mm column and 5 μm particles (Supelco). The Eluent A was THF/ACN/5 mM Ammonium formate (3/2/5 v/v/v) with 0.1% formic acid and eluent B was THF/ACN/5 mM Ammonium formate (7/2/1 v/v/v) with 0.1% formic acid. The gradient elution program for Cer and GluCer quantification was as follows: 0–1 min, 1% eluent B; 40 min, 80% eluent B; and 40–42, 80% eluent B. The gradient elution program for GIPC quantification was as follows: 0–1 min, 15% eluent B; 31 min, 45% eluent B; 47.5 min, 70% eluent B; and 47.5–49, 70% eluent B. The flow rate was set at 0.2 mL/min, and 5 mL sample volumes were injected. The areas of LC peaks were determined using MultiQuant software (ABSciex) for sphingolipids relative quantification to the area of the internal standard normalized to the fresh weight, values are expressed in arbitrary unit (AU).

**Label-free LC-MS/MS proteomic analysis of SVs/TGN compartments.** Proteins of the IP output fraction were eluted by adding 25 μL of 1% SDS, 0.3 μL of 2 M dithiothreitol, 2.3 μL of 1 M iodoacetamide, and 6.9 μL of 5× Laemmli buffer sequentially (the volume of each reagent is for 75 μL Dynabeads in initial amount) with an incubation for 30 min at 37 °C (except for iodoacetamide at the room temperature) between each addition. The protein amounts of the eluted samples were equilibrated using the Stain-Free protein visualization system similarly to the loading controls for western blotting described above. Samples from four biological replicates were used for quantification. The equilibrated samples were solubilized in Laemmli buffer and deposited onto SDS-PAGE gel for concentration and cleaning purposes. After colloidal blue staining, bands were cut out from the gel and subsequently cut into 1 mm$^3$ pieces. Gel pieces were destained in 25 mM ammonium bicarbonate and 50% acetonitrile (ACN), rinsed twice in ultrapure water, and shrunk in ACN for 10 min. After ACN removal, the gel pieces were dried at room temperature, covered with trypsin solution (10 ng/μL in 50 mM NH$_4$HCO$_3$), rehydrated at 4 °C for 10 min, and finally incubated overnight at 37 °C. Gel pieces were then incubated for 15 min in 50 mM NH$_4$HCO$_3$ at room temperature with rotary shaking. The supernatant was collected, and an H$_2$O/ACN/HCOOH (47.5:47.5:5) extraction solution was added onto gel slices for 15 min. The extraction step was repeated twice. Supernatants were pooled and concentrated in a vacuum centrifuge to a final volume of 100 μL. Digests were finally acidified by addition of 2.4 μL of formic acid (5% v/v).

Peptide mixture was analyzed on an Ultimate 3000 nanoLC system (Dionex) coupled to an Electrospray Q-Exactive quadrupole Orbitrap benchtop mass spectrometer (Thermo Fisher Scientific). Ten microliters of peptide digests were loaded onto a C18 PepMap trap column (300 μm inner diameter × 5 mm, Thermo Fisher Scientific) at a flow rate of 30 μL/min. The peptides were eluted from the trap column onto an analytical C18 PepMap column (75 μm inner diameter × 25 cm, Thermo Fisher Scientific) with a 4–40% linear gradient of solvent B in 108 min (solvent A was 0.1% formic acid in 5% ACN, and solvent B was 0.1% formic acid in 80% ACN). The separation flow rate was set at 300 nL/min. The mass spectrometer was operated in positive ion mode at a 1.8 kV needle voltage. Data were acquired using Xcalibur 2.2 software in a data-dependent mode. MS scans (m/z 350–1600) were recorded at a resolution of R = 70,000 (m/z 200) and an AGC target of 3 ×

$10^6$ ions collected within 100 ms. Dynamic exclusion was set to 30 s and top 12 ions were selected from fragmentation in HCD mod. MS/MS scans with a target value of $1 \times 10^5$ ions were collected with a maximum fill time of 100 ms and a resolution of R = 175,000. Additionally, only +2 and +3 charged ions were selected for fragmentation. The other settings were as follows: no sheath nor auxiliary gas flow, heated capillary temperature at 250 °C, normalized HCD collision energy of 25%, and an isolation width of 2 $m/z$.

Data were searched by SEQUEST through Proteome Discover (Thermo Fisher Scientific) against Araport v11 protein database. Spectra from peptides higher than 5000 Da or lower than 350 Da were rejected. The search parameters were as follows: mass accuracy of the monoisotopic peptide precursor and peptide fragments was set to 10 ppm and 0.02 Da respectively. Only b- and y-ions were considered for mass calculation. Oxidation of methionines (+16 Da) was considered as variable modification and carbamidomethylation of cysteines (+57 Da) as fixed modification. Two missed trypsin cleavages were allowed. Peptide validation was performed using Percolator algorithm[77] and only "high confidence" peptides were retained corresponding to a 1% False Positive Rate at peptide level.

For label-free quantitative data analysis, raw LC-MS/MS data were imported in Progenesis QI for Proteomics (Nonlinear Dynamics). Data processing includes the following steps: (i) features detection. (ii) features alignment across the samples to compare, (iii) volume integration for 2–6 charge-state ions, (iv) normalization on features ratio median, (v) import of sequence information, (vi) calculation of protein abundance (sum of the volume of corresponding peptides), (vii) a Wilcox rank-sum test to compare each group and filtering of proteins based on $p$-value < 0.05. Only nonconflicting features and unique peptides were considered for calculation at protein level.

The mass spectrometry proteomics data have been deposited to the ProteomeXchange Consortium via the PRIDE[78] partner repository with the dataset identifier PXD026252.

**Statistics**. For the comparison of two groups, two-sided Wilcoxon's rank-sum test was used. Kruskal–Wallis test followed by Dwass-Steel-Critchlow-Flinger multiple comparison test was used for the comparison of 3 and more groups. All the statistics were performed with R (version 3.6.0) and RStudio (version 1.2.1335). Variances between each group of data are represented in boxplot, bee swarm or dotplot. Each element of the boxplot indicates the following value: center line, median; box limits, the first and third quartiles; whiskers, 1.5× interquartile range; points above or below the whiskers, outliers. $P$-values are described in supplementary data 1. Sample sizes are described in figure legends.

**Reporting summary**. Further information on research design is available in the Nature Research Reporting Summary linked to this article.

## Data availability

Data supporting the findings of this work are available within the paper and its Supplementary Information files. The mass spectrometry proteomics data have been deposited to the ProteomeXchange Consortium via the PRIDE partner repository with the dataset identifier PXD026252. All other datasets and plant materials generated and analyzed during the current study are available from the corresponding author upon request. Source data are provided with this paper.

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

## Acknowledgements

We are grateful to Sebastien Mongrand (LBM, Bordeaux, France) and Emmanuelle Bayer (LBM, Bordeaux, France) for critical reading of the manuscript. We thank Fabrice Cordelières (Bordeaux Neurocampus, France) for discussions and advices on microscopy images quantification and Clément-Marie Train (Swiss Institute of Bioinformatic, Lausanne, Switzerland) for his help on proteomic data. Lipidomic analyses were performed on the Bordeaux Metabolome Facility-MetaboHUB (ANR-11-INBS-0010). Imaging was performed on the plant unit of the Bordeaux Imaging Center (BIC), part of the National Infrastructure France-BioImaging supported by the French National Research Agency (ANR-10-INSB-04). This work was supported by a PhD funding from the French ministry of research (MENRT) granted to Y.B. to support the PhD thesis of N.E, Overseas Research Fellowship granted from Japan Society for Promotion of Science (JSPS) to Y.I., a research grant from the French National Research Agency to Y.B. and Y.J. (ANR-18-CE13-0025), and a research grant from the European Research Council (ERC) to Y.J. (336360-APPL).

## Author contributions

Y.B. conceptualized and designed the experiments with input from P.M. and Y.J. Y.I. performed experiments in Figs. 2, 4, 5, 7, 9, 10 and supplementary Fig. 1, 3, 4, 6, 9. N.E. performed experiments in Fig. 1 and generated materials for proteomic experiment in Fig. 8 and supplementary Fig. 8, S.C. performed LC-MS proteomics of Fig. 8 and supplementary Fig. 8, Y.I., N.E, S.C. and Y.B. analyzed proteomic results. Y.B., W.M., and L.F. performed lipid extraction and analysis in Figs. 3, 4, 6 and supplementary Fig. 2, 4, 5, L.F. performed LC-MS/MS experiments and analyzed the results. V.W.-B. performed experiments in Fig. 7 and supplementary Fig. 7. M.P. and L.N. provided crossed lines and plant materials crucial for this study. All quantifications and statistics were performed by Y.I. Y.I. built all the figures. Y.B. wrote the manuscript with input from all the authors. Y.B. acquired funding and supervised all aspect of the study. All authors reviewed, edited, and approved the manuscript.

## Competing interests

The authors declare no competing interests.

**Additional information**

