## [Peer Review File · Nature Communications]

REVIEWER COMMENTS

Reviewer #1 (Remarks to the Author):

Review of "Sphingolipid mediate polar sorting of PIN2 through phosphoinositide consumption at the trans-Golgi Network"

In this manuscript the authors claim that reducing the chain length of sphingolipids by incubation with the drug metazachlor causes an increase in the PI4p at the trans-Golgi and therefore a disruption in the polar secretion of PIN2. While this conclusion is consistent with their results, the entire demonstration relies on inhibitors, whose specificities have not been entirely established and techniques for quantification of PIPs and proteins at the TGN that are questionable. So, while the mechanism would be interesting if it were firmly established, my enthusiasm is tempered by the number of issues with the analysis. I describe below some of my reservations.

1. Metazachlor. What is the evidence that metazachlor only effects hydroxylated fatty acid elongation? Normally, to show something like this, for a particular phenomenon, they would need to treat with the inhibitor and add back C24 and C26 hydroxylated fatty acids. Roots might be able to take them up. Other ways are available to get fatty acids into cells, for instance using caged compounds, but these require quite a bit of work and as far as I know have not been used much in plants. I am not aware that the add back experiment has been done in plants or even in plant tissue culture. Without a good control for specificity, an alternative would be to use genetic methods. For example, they could intervene genetically on the target of metazachlor. Of course, this is likely to be problematic because there seems to be developmental problems without very long chain sphingolipids. I am not that familiar with the possibilities of regulated promoters in plants, but if their story would be more convincing with an alternative to the drug. Another point about metazachlor is that they say that it only affects sphingolipids almost throughout the paper, even though they have shown previously that it also affects very long chain phospholipids, which are just present in lower amounts. Finally they address this in the discussion and argue it away, but they really don't have convincing results that this is due to sphingolipids and not some other very long chain lipid molecule. One way again, would need regulated expression of ceramide synthases LOH1 and LOH3. I don't know if this is feasible in plants, it would be in animal cells or yeast, for example with an inducible degron or regulated expression. First, I don't understand why they haven't discussed the paper from Markham (Plant Cell, 23 (6) (2011), pp. 2362-2378) that shows that the very long chain sphingolipids are required for polar sorting of PIN2. This clearly sets a precedent for their finding. Manipulation of the ceramide synthases that use the very long fatty acids would eliminate the issue with the phospholipids, although it may result in their increase.

2. Use of the fluorescent probes for PIP quantitation. The probes used for PIP quantification are widely used and often misused. In fact, there is substantial evidence that their labeling is context dependent. That is, the intensity of the signal may depend upon the lipid environment of the PIP and also on the availability of competing PIP binding proteins in the cytosol. For example, if a prominent-PIP binding protein is sequestered elsewhere than where they are looking at there will be less competition and a higher labeling. To completely rely on them for quantitation, rather than simply localization, is a stretch. They need to provide evidence that the probes are quantitative in vivo or use alternative methods to confirm their results. They are able to isolate TGN. They could measure the PIPs in such preps.

3. Label free proteomics. They do quite elegant immunoprecipitation and label free proteomics on TGN fractions. They admit the signal is not necessarily representative of absolute quantities and then they go on to discuss the data ignoring this. Nevertheless, the decrease in PI-PLC seems to be large. What is the statistical significance of this difference and can they confirm the reduction at the TGN by another method, such as fluorescence microscopy? This should be doable and would increase the confidence in their data. This is a crucial finding for their mechanism and it is important to validate the proteomics data, especially since the TGN is not pure (they did a great job on it, but there are still endosomes and traces of other compartments).

4. The same points hold for the PI-PLC inhibitor. How specific is it? Can they find an alternative method to confirm this result. In this case, I don't see how they could do an add back experiment as with the very long chain fatty acids.

In the writing of the manuscript, especially the introduction, the authors should better separate facts from hypotheses. As an example "SL and sterols are enriched at the TGN where they

increase membrane thickness and assemble to form small sorting platforms, both of which are important for protein sorting and trafficking". As far as I know much of this is speculation. These two lipids seem to be enriched at the TGN. In artificial membranes they can increase membrane thickness, but they have never been shown to do this with the lipid and protein composition of the TGN. It could be, for example, that membrane protein composition plays a predominant role in membrane thickness. Neither have membrane thickness or sorting platforms ever been shown to play a specific role in protein sorting. It is not a problem to repeat some of these talking points in the literature, but they should not be accepted without proof. This was just one example. Please be more precise describing what is shown and what is suggested.

Finally, I realize that I am being quite demanding in this review and some of these suggestions might be very difficult to carry out practically, but their conclusions cannot be so certain without additional information and alternative methods. An alternative to reaching completely solid conclusion about the whole mechanism would be to recognize the shortcomings of the approaches, address them where possible and leave some things open where the evidence is not tight.
Howard Riezman

Reviewer #2 (Remarks to the Author):

Here the authors mainly use metazachlor to affect the acyl chain length of sphingolipids (SL) and observe a correlation with altered intensity of PI4P sensors. The authors conclude that PI4P levels at the TGN are altered in a SL-dependent manner. The authors may discuss the limitations of such biosensors, because it could be that not the PI4P levels but simply the availability for sensor binding (which may compete with endogenous binding proteins) is altered. The authors suggest that TGN residency of certain PI-PLCs may be sensitive to metazachlor and PI-PLC inhibitor treatments suggest an involvement in regulating PI4Ps availability at the TGN. The work is of high quality and I have only one major concern. The work is mainly based on inhibitor, such as metazachlor and U73122 as well as their co-treatments treatments. It would have been nice to substantiate some of the findings using genetic interference (especially the genetic dissection of PI-PLC's role would have been nice).

Minor points that the author may want to address:

The depiction of PI-PLCs localization at the TGN and its mis-localization after metazachlor would have been informative to support the proteomic approach.

mCIT-1xPHFAPP1-E50A and mCIT-1xPHFAPP1-E50A-H54A intensity at the PM were not decreased upon metazachlor, but mCIT-1xPHFAPP1 sensor intensity at the PM was sensitive. Could the author, please, better explain this difference.

The authors immunopurified SYP61-SVs/TGN compartments (Fig. 2d) and characterized the fatty acid (FA) and sterol composition by GC-MS in untreated and metazachlor-treated seedlings and did not detect significant alterations in sterol content. I am not sure if this negative result strongly argues against a sterol exchange mechanism. Besides technical/sensitivity issues, there could be a mechanism in place that balances the sterol concentrations after the exchange mechanism. The discussion could be more balanced here.

Would available pi4 kinase mutants have altered sensitivity when grown on metazachlor?

Reviewer #1 (Remarks to the Author):

Review of “Sphingolipid mediate polar sorting of PIN2 through phosphoinositide consumption at the trans-Golgi Network”

In this manuscript the authors claim that reducing the chain length of sphingolipids by incubation with the drug metazachlor causes an increase in the PI4p at the trans-Golgi and therefore a disruption in the polar secretion of PIN2. While this conclusion is consistent with their results, the entire demonstration relies on inhibitors, whose specificities have not been entirely established and techniques for quantification of PIPs and proteins at the TGN that are questionable. So, while the mechanism would be interesting if it were firmly established, my enthusiasm is tempered by the number of issues with the analysis. I describe below some of my reservations.

We would like to thank Professor Howard Riezman for his positive and helpful comments. We basically took them all in consideration and we performed new experiments to address these comments, mostly by implementing genetic tools to confirm our inhibitors results, setting up add-back experiments, performing lipidomics of immuno-purified TGN and sphingolipid mutants, and finally by extending our cell biology analyses. The manuscript is now substantially revised; we hope this version of the manuscript will sufficiently address the issues raised.

1. Metazachlor. What is the evidence that metazachlor only effects hydroxylated fatty acid elongation?

*Our previous work showed that metazachlor does not only alter α -hydroxylated very-long chain fatty acids (hVLCFAs) but also non- α -hydroxylated VLCFAs (Wattelet-Boyer et al., 2016). The common feature is that metazachlor alters the elongation step specifically between 22 and 24 carbons. However, previously we have shown that immuno-purified TGN compartments are enriched in hVLCFAs, mostly h24 and h26, but not non- α -hydroxylated VLCFAs (Wattelet-Boyer et al., 2016). hVLCFAs are specific to sphingolipids (SLs) while VLCFAs of phospholipids are non- α -hydroxylated only. VLCFAs constitute about 85% and 2% of the pools of sphingolipids and phospholipids respectively in *Arabidopsis* root (Wattelet-Boyer et al., 2016). Hence, the effect of metazachlor in the pool of SLs is drastic while it remains minor in the pool of phospholipids (Wattelet-Boyer et al., 2016). However, we completely agree that this does not sufficiently supports the conclusion that the effect of metazachlor at the cellular level is mediated by SLs. In principle, metazachlor could impact all VLCFA-containing membrane lipids such as SLs but also the phospholipid PS which is by far the major phospholipid with VLCFA in plants (Murata et al., 1984; Platre et al., 2018).*

To address this point we used the add-back and genetic approaches that were suggested below.

Normally, to show something like this, for a particular phenomenon, they would need to treat with the inhibitor and add back C24 and C26 hydroxylated fatty acids. Roots might be able to take them up. Other ways are available to get fatty acids into cells, for instance using caged compounds, but these require quite a bit of work and as far as I know have not been used much in plants. I am not aware that the add back experiment has been done in plants or even in plant tissue culture.

*To our knowledge, add-back experiments with VLCFAs have only been performed in plants in one paper on the role of VLCFAs in cotton fiber and *Arabidopsis* cell elongation (Qin et al., 2007). Both non- α -hydroxylated 24:0 and α -hydroxylated h24:0 are commercially available, so we took this opportunity to try add-back of these VLCFAs in metazachlor-treated seedlings. Contentedly, it*

worked. Our results show that plants are able to take them up, fatty acids analyses in metazachlor-treated roots fed-back with either 24:0 or h24:0, showed restored level of these VLCFAs. Consistently with our hypothesis that the enrichment of hVLCFAs at TGN is involved in the cellular effects we describe, only h24:0, but not 24:0, was able to prevent the accumulation of PI4P at the TGN upon metazachlor. Thus, although metazachlor decreases both 24:0 and h24:0, only h24:0 is functionally relevant for the effects we observed at the TGN. We do not exclude that 24:0 mediates other processes, this is undoubtedly the case, but for our present study h24:0, not 24:0, is involved in PI4P homeostasis at the TGN. Moreover, we do not exclude that h24:1 is also involved in PI4P homeostasis at TGN but we could not test this as h24:1 is not commercially available. Given that h24:0 is specific to SLs these results suggested that SLs, but not the phospholipid pool, are involved in PI4P homeostasis at the TGN. To test this further, we used SLs and PS mutants (described in the points below). In the future, we will be interested in exploiting the possibility of using caged lipids to address more precisely the function and/or localization of SLs at TGN.

Without a good control for specificity, an alternative would be to use genetic methods. For example, they could intervene genetically on the target of metazachlor. Of course, this is likely to be problematic because there seems to be developmental problems without very long chain sphingolipids. I am not that familiar with the possibilities of regulated promoters in plants, but it their story would be more convincing with an alternative to the drug.

The target of metazachlor is the keto-acyl synthase (KCS) protein family (Tresch et al., 2012). In Arabidopsis, the KCS family includes 21 proteins spread into several phylogenetic clades (Joubès et al., 2008). Expression of Arabidopsis KCS genes in yeast revealed that metazachlor could potentially targets at least 7 KCS in Arabidopsis seedlings (Tresch et al., 2012). Thus, genetic approaches on KCS often face gene redundancy issues. However, in our previous work we could identify that KCS2, KCS20 and KCS9 were functionally relevant for PIN2 sorting at the TGN and the phenotypic readout we used in that study, i.e. the response of the main root to gravity (gravitropism) (Wattelet-Boyer et al., 2016). To uncover the involvement of KCSs in VLCFA-mediated root gravitropism, we previously used a concentration range of metazachlor and found a concentration that induced a phenotypic response in kcs mutants but not in the wild-type. This indicated a hypersensitivity of the kcs mutants to metazachlor, which was expected since KCSs are the biochemical targets of metazachlor (Wattelet-Boyer et al., 2016). KCS2 and KCS20 need to be combined to double mutant in order to visualize phenotypic defects while KCS9 is displaying some phenotypes as single mutant (Kim et al., 2013; Lee et al., 2009). We therefore looked at kcs9 single mutant and investigated whether it would display hypersensitivity to metazachlor with respect to PI4P accumulation at the TGN. We identified a concentration of metazachlor for which we could detect a small but significant increase of PI4P at the TGN, compared to non-treated plants, while this PI4P increase was not observed in Col-0 wild-type (treated with the same concentration of metazachlor). These genetic data suggest that KCS9 and metazachlor act in one pathway to exert their effects on PI4P at the TGN.

Another point about metazachlor is that they say that it only affects sphingolipids almost throughout the paper, even though they have shown previously that it also affects very long chain phospholipids, which are just present in lower amounts. Finally, they address this in the discussion and argue it away, but they really don't have convincing results that this is due to sphingolipids and not some other very long chain lipid molecule.

*We fully agree. Although minor, the 2% of VLCFAs in the phospholipid pool could play a role in the process we describe. To address this question, the add-back experiments with h24:0 and 24:0 (described in the point above) is a good first answer as only h24:0, which is specific of SLs, is able to rescue PI4P accumulation at the TGN. PS is by far the major phospholipid with VLCFA in plants (Murata et al., 1984; Platre et al., 2018). To get further genetic insights into this question we analyzed PI4P distribution in the knock-out PS synthase mutant *pss1* which does not produce any PS (Platre et al., 2018) as well as in a knock-down mutant of *IPCS*, the enzyme that grafts an inositolphosphate group on ceramide to produce inositolphosphoryl ceramide (IPC), which is further glycosylated to produce the most abundant form of SLs in plants, the glycosylinositolphosphoryl ceramides (GIPCs).*

*In the *pss1* mutant we did not observe any changes in PI4P subcellular patterning, neither in intracellular structures nor at the PM. These new results show that PS does not impact PI4P homeostasis.*

*On the SLs side, we focused on *IPCS* enzymes given that they produce GIPC from ceramides and that *IPCS2* was proposed to localize at the TGN (Wang et al., 2008). *IPCS* enzymes family contains 3 members amongst which *IPCS2* is the main expressed isoform in *Arabidopsis* root with around 50 fragments per kilo base per million mapped reads (FPKM) found in RNA-seq database while *IPCS1* and *IPCS3* score around 25 FPKM and 10 FPKM, respectively, in *Arabidopsis* root. First, we precisely localized *IPCS2* using a functional *IPCS2*-tagRFP construct driven by its own promoter. Interestingly, we could observe that *IPCS2* localizes in intracellular dots that strongly co-localize with a marker that labels the secretory vesicles (SVs) of the TGN while the co-localization with other markers that either label the clathrin-coated vesicles (CCVs) of the TGN or the Golgi apparatus display weak co-localization. These results are in perfect agreement with our previous finding that the SVs sub-domain of the TGN is enriched in hVLCFAs, that are specific of SLs, as compared to either CCVs of the TGN or Golgi (Wattelet-Boyer et al., 2016). Furthermore, we generated an *ipcs1;ipcs2* double knock-out mutant which displays severe developmental phenotype with early growth arrest. We used this double mutant to show that our *pIPCS2::IPCS2*-tagRFP construct was functional and could rescue the mutant phenotype but did not use it for further analyses given the strong phenotype. Alternatively, we produced an estradiol-inducible artificial microRNA (*amiRNA*) line producing *amiRNA* directed against *IPCS2* and *IPCS1* (*ipcs1;ipcs2^{amiRNA}*). Upon induction, the root growth was reduced but the overall phenotype was much less drastic and hence usable. We performed sphingolipidomics on root samples of *ipcs1;ipcs2^{amiRNA}* line and could quantify that the level of VLCFA-glycosylated SLs (VLCFA-GIPC or VLCFA-glucosylceramide) was decreased while the level of ceramides was increased, consistently with the function of *IPCS* enzymes. Finally, we introduced a PI4P marker into this genetic background and could observe an increase of PI4P at the TGN indicating that SLs are involved in PI4P homeostasis at TGN.*

One way again, would need regulated expression of ceramide synthases LOH1 and LOH3. I don't know if this is feasible in plants, it would be in animal cells or yeast, for example with an inducible degen or regulated expression.

*We used inducible *amiRNA* directed against *IPCS1* and *IPCS2* and observed PI4P accumulation at the TGN. However, given that the level of ceramides increases in *ipcs1;ipcs2^{amiRNA}* line we wanted to test whether ceramide composition could impact PI4P homeostasis at TGN.*

To address this question we did not use loh1;loh3 double mutant as this mutant also displays a drastic developmental phenotype, instead we used the drug fumonisinB1 (FB1). Using this experimental condition, we could quantify that FB1 treatment does not alter the total quantity of ceramides but rather alters the composition of fatty acids within the ceramide pool by reducing VLCFA-ceramides and increasing C16-ceramides. With FB1 we could detect a slight but significant increase of PI4P at the TGN confirming by yet another independent approach that SLs are involved in PI4P homeostasis at TGN. However, we also noticed that metazachlor treatment impact more strongly PI4P homeostasis at TGN than FB1 treatment. These results may indicate that VLCFA-ceramides have a rather small effect on PI4P homeostasis at the TGN and that VLCFA-glycosylated SLs such as GIPC could be instead involved in this process. We do not extensively discuss this point as our data do not allow to address it further. In any case, our results now show that SLs are involved in PI4P homeostasis at the TGN.

First, I don't understand why they haven't discussed the paper from Markham (Plant Cell, 23 (6) (2011), pp. 2362-2378) that shows that the very long chain sphingolipids are required for polar sorting of PIN2. This clearly sets a precedent for their finding. Manipulation of the ceramide synthases that use the very long fatty acids would eliminate the issue with the phospholipids, although it may result in their increase.

The study from Markham et al., 2011 shows that VLCFA of ceramides play a role in the polar localization of the auxin carrier PIN1, but not PIN2 (Markham et al., 2011). This is quite important since PIN1 and PIN2 localize at distinct polar domain of the plasma membrane (PM), PIN1 is polar at the basal membrane while PIN2 is polar at the apical membrane. Moreover, it was shown previously that their traffic is regulated by distinct mechanisms and pathways (Kleine-Vehn et al., 2008). Actually, in Markham et al., 2011, PIN1 and AUX1 trafficking was altered but not PIN2 while we show in Wattelet-Boyer et al., 2016 that metazachlor alters the polar sorting of PIN2 but not of PIN1 nor AUX1. Moreover, Markham et al., 2011 have shown that ceramides-dependent PIN1 mis-sorting occurs at RAB-A2a (Rab11 in animals, Ypt3 in yeasts) compartments, which we identified as being TGN-associated CCVs in Wattelet-Boyer et al., 2016, while metazachlor-induced mis-sorting of PIN2 occurs at TGN-associated SVs, not CCVs. Given our results, we guess that SVs/TGN and CCVs/TGN do not have the same SL composition (ceramides vs glycosylated SLs) and that the different SL species present in these sub-domains of TGN would fulfil distinct function during protein sorting. Consistently with this hypothesis, we find in this paper that IPCS localizes at SVs/TGN (while being weak at CCVs). Targeted localization of enzymes that glycosylate ceramides could be a way for plant cells to segregate trafficking pathways and sorting machineries according to distinct sphingolipid signatures. We discussed these aspects into a recent review in FEBS "special issue on sphingolipids biology" (Mamode Cassim et al., 2020). This hypothesis on differential SL composition at TGN sub-domains is a real interesting twist for future research in the field of protein sorting. We introduced a paragraph on this in the discussion of the revised manuscript.

2. Use of the fluorescent probes for PIP quantitation. The probes used for PIP quantification are widely used and often misused. In fact, there is substantial evidence that their labeling is context dependent. That is, the intensity of the signal may depend upon the lipid environment of the PIP and also on the availability of competing PIP binding proteins in the cytosol. For example, if a prominent-PIP binding protein is sequestered elsewhere than where they are looking at there will be less competition and a higher labeling. To completely rely on them for quantitation, rather than simply localization, is a stretch. They need to provide evidence that the probes are quantitative in vivo or

use alternative methods to confirm their results. They are able to isolate TGN. They could measure the PIPs in such preps.

We agree that PIPs biosensors might not completely reveal all the pools of PIPs present in the cell, depending on their avidity for the PIPs, competition, turnover at membranes. As a note, in this work we rely on several PIPs biosensors that all show the same result when we applied metazachlor, i.e. an increased level of the probe at the TGN. Taking the reviewer comment into consideration and to confirm our findings we performed TGN immuno-precipitation and quantified the level of PIPs in these fractions by LC-MS/MS. In fact, PIPs identification and quantification by LC-MS/MS is a routinely used method in animal cells (Clark et al., 2011) but has never been performed in plants before. So far, in plants, PIPs quantification has been performed by using thin layer and gas chromatography (Heilmann and Heilmann, 2013). However, due to the limited amount of material we can get from an IP we decided not to implement the HPTLC/GC-MS approach on our samples but rather to adapt the protocol described in Clark et al., 2011. Fortunately, we were successful in extracting PIPs from our IP samples and in identifying and quantifying them by LC-MS/MS. Our results revealed an increased level of all PIPs species in the TGN IPs from metazachlor-treated seedlings as compared to non-treated seedlings. These results are indeed an important confirmation of what we described by the use of the biosensors.

3. Label free proteomics. They do quite elegant immunoisolation and label free proteomics on TGN fractions. They admit the signal is not necessarily representative of absolute quantities and then they go on to discuss the data ignoring this. Nevertheless, the decrease in PI-PLC seems to be large. What is the statistical significance of this difference and can they confirm the reduction at the TGN by another method, such as fluorescence microscopy? This should be doable and would increase the confidence in their data. This is a crucial finding for their mechanism and it is important to validate the proteomics data, especially since the TGN is not pure (they did a great job on it, but there are still endosomes and traces of other compartments).

We fundamentally agree. What we stated in the text is that label free proteomics aims at comparing protein abundancies of a given sample upon different conditions. Thus, label free is an accurate method to compare the abundance of a given protein between metazachlor-treated TGN IP and untreated TGN IP. We were more careful in comparing different proteins within either metazachlor-treated TGN IPs or untreated TGN IPs. We agree that TGN IPs cannot be completely 100% pure. Indeed, PM proteins were detected in our TGN IPs but we cannot discriminate whether they are contaminants from the PM or whether they are in transit through the TGN to reach the PM. Hence, we agree that some biases should be considered especially in the case of PI-PLCs which mainly reside at the PM (Kanehara et al., 2015; Otterhag et al., 2001). However, in our previous study we checked protein markers of other compartments by western-blot and could confirm that the SYP61 TGN IP fraction is highly enriched for TGN proteins and depleted for either Golgi or more importantly PM proteins (Wattelet-Boyer et al., 2016). A point to consider is that we identified a good number of characterized TGN-localized proteins in our IPs and that some of them are amongst the most abundant proteins we found. Moreover, in our proteomics we found only very low abundance of either Golgi or multivesicular bodies markers further confirming the high purity of our IPs. Importantly, the abundance of the TGN-localized proteins was not altered upon metazachlor meaning that TGN still keeps its identity upon metazachlor treatment.

To calculate the statistical significance of the differences found, we applied a Wilcox rank sum test to filter proteins based on p-value <0.05. We included these p-values in the new versions of supplementary Data 1 and 2. According to statistics, we now detected a significant decrease

(minimum 2-fold threshold) of 189 proteins, including PI-PLCs, and a significant increase (minimum 2-fold threshold) of 127 proteins at TGN upon metazachlor. These results indicate that out of 4458 proteins the quantity of only 316 proteins were altered upon metazachlor treatment. Thus, the altered abundance of proteins upon metazachlor represents only 7% of the proteins found at the TGN, which looks like a rather focused population of proteins targeted.

Using cell fractionation, previous studies have suggested that PI-PLCs could locate not only at the PM but as well in endomembrane compartments (Otterhag et al., 2001). We had a careful look at PI-PLC2-venus construct driven by its own promoter (Kanehara et al., 2015) but could not detect any intracellular compartments labelled. However, the signal intensity in that line was extremely weak even at the PM. Thus, we decided to generate a PI-PLC2-GFP line driven by a stronger promoter, the Ubiquitin10 promoter which is used in plants to get a reasonable good level of expression (Geldner et al., 2009). Our pUBQ10::PLC2-GFP line showed a strong labeling at PM but also a weak labeling in intracellular structures. The difference of intensity between PM and TGN complicated the imaging and quantification. To tackle this issue, we had to apply a Gaussian filter and subtract the background, in these conditions we could detect intracellular structures that partly co-localized with a TGN marker. This result confirmed that PI-PLC2 localizes at the TGN additionally to the PM. Unfortunately, due to the low signal of PI-PLC2 at the TGN we could not quantify the intensity of fluorescence at the TGN in either control or metazachlor-treated seedlings. However, we are confident that PI-PLCs impact PI4P homeostasis at TGN given that we quantified an increase of PI4P at the TGN upon PI-PLC inhibitor or in the pi-plc2 knock-out mutant. Our data with the PI-PLC inhibitor revealed the PI4P increase at TGN cannot be increased by addition of metazachlor, suggesting that VLCFAs of SLs and PI-PLCs act in one pathway.

4. The same points hold for the PI-PLC inhibitor. How specific is it? Can they find an alternative method to confirm this result. In this case, I don't see how they could do an add back experiment as with the very long chain fatty acids.

We agree that although U-73122 has been widely used as a potent PI-PLC inhibitor, a genetic alternative would greatly backup our data. In Arabidopsis, PI-PLCs is a family of 9 members (Pokotylo et al., 2014). We choose to focus on PI-PLC2 as it is the main expressed isoform expressed in the root and our proteomics identified PLC2 and/or PLC7 as being decreased upon metazachlor treatment. PLC2 mutants are not easy to work with as they display gametophyte lethality (Chen et al., 2019). A weak mutant allele exists but it is not in the same genetic background as the PIP biosensors and displays weak phenotypes (Kanehara et al., 2015). A strong allele is in Col-0 background but displays strong gametophyte lethality, the homozygote seedlings can grow but no seeds can be rescued from them (Chen et al., 2019). Still, we introduced a PIP biosensor into this genetic background and screened a high number of seedlings in the segregating population (that contains heterozygotes, wild-type and a low proportion of homozygous mutants) to select homozygote seedlings expressing the biosensor. In this background, we did not find 25% of homozygotes but rather around 5% which confirmed the gametophyte lethality of pi-plc2 mutant. Using a PI4P biosensor, our results show that the PI4P level increases at the TGN as well as at the PM in the pi-plc2 mutant background, consistently with our findings on the localization of PLC2. These results add a genetic confirmation to our findings.

In the writing of the manuscript, especially the introduction, the authors should better separate facts from hypotheses. As an example “SL and sterols are enriched at the TGN where they increase membrane thickness and assemble to form small sorting platforms, both of which are important for protein sorting and trafficking”. As far as I know much of this is speculation. These two lipids seem to be enriched at the TGN. In artificial membranes they can increase membrane thickness, but they have never been shown to do this with the lipid and protein composition of the TGN. It could be, for example, that membrane protein composition plays a predominant role in membrane thickness. Neither have membrane thickness or sorting platforms ever been shown to play a specific role in protein sorting. It is not a problem to repeat some of these talking points in the literature, but they should not be accepted without proof. This was just one example. Please be more precise describing what is shown and what is suggested.

Yes we agree, the sentence was removed to keep only “SL and sterols are enriched at the TGN and are important for protein sorting and trafficking” as our study do not make any point on a potential function of SL/sterols platforms in protein sorting. Indeed, we agree that this is a major issue that needs to be addressed in the future in the membrane biology field.

Finally, I realize that I am being quite demanding in this review and some of these suggestions might be very difficult to carry out practically, but their conclusions cannot be so certain without additional information and alternative methods. An alternative to reaching completely solid conclusion about the whole mechanism would be to recognize the shortcomings of the approaches, address them where possible and leave some things open where the evidence is not tight.
Howard Riezman

We would like to thanks again Howard Riezman for his comments that made our study much more solid by implementing genetic approaches as well as alternative methods for our main claims. We also revised the text to be more cautious on our interpretations.

Reviewer #2 (Remarks to the Author):

Here the authors mainly use metazachlor to affect the acyl chain length of sphingolipids (SL) and observe a correlation with altered intensity of PI4P sensors. The authors conclude that PI4P levels at the TGN are altered in a SL-dependent manner.

We would like to thank reviewer 2 for his/her constructive comments on our manuscript. We performed new experiments to address these comments, mostly by implementing genetic tools to confirm our inhibitors results, performing lipidomics of immuno-purified TGN and sphingolipid mutants, and finally by extending our cell biology analyses. The manuscript is now substantially revised; we hope this version of the manuscript will sufficiently address the issues raised.

The authors may discuss the limitations of such biosensors, because it could be that not the PI4P levels but simply the availability for sensor binding (which may compete with endogenous binding proteins) is altered.

We agree that PIPs biosensors might not completely reveal all the pools of PIPs present in the cell, depending on their avidity for the PIPs, competition, turnover at membranes. As a note, in this work we rely on several PIPs biosensors that all show the same result when we applied metazachlor, i.e. an increased level of the probe at the TGN. Taking the reviewer comment into consideration and to confirm our findings we performed TGN immuno-precipitation and quantified the level of PIPs in these fractions by LC-MS/MS. In fact, PIPs identification and quantification by LC-MS/MS is a routinely used method in animal cells (Clark et al., 2011) but has never been performed in plants before. So far, in plants, PIPs quantification has been performed by using thin layer and gas chromatography (Heilmann and Heilmann, 2013). However, due to the limited amount of material we can get from an IP we decided not to implement the HPTLC/GC-MS approach on our samples but rather to adapt the protocol described in Clark et al., 2011. Fortunately, we were successful in extracting PIPs from our IP samples and in identifying and quantifying them by LC-MS/MS. Our results revealed an increased level of all PIPs species in the TGN IPs from metazachlor-treated seedlings as compared to non-treated seedlings. These results are indeed an important confirmation of what we described by the use of the biosensors.

The authors suggest that TGN residency of certain PI-PLCs may be sensitive to metazachlor and PI-PLC inhibitor treatments suggest an involvement in regulating PI4Ps availability at the TGN. The work is of high quality and I have only one major concern. The work is mainly based on inhibitor, such as metazachlor and U73122 as well as their co-treatments treatments. It would have been nice to substantiate some of the findings using genetic interference (especially the genetic dissection of PI-PLC's role would have been nice).

We agree that although U-73122 has been widely used as a potent PI-PLC inhibitor, a genetic alternative would greatly backup our data. In Arabidopsis, PI-PLCs is a family of 9 members (Pokotylo et al., 2014). We choose to focus on PI-PLC2 as it is the main expressed isoform expressed in the root and our proteomics identified PLC2 and/or PLC7 as being decreased upon metazachlor treatment. PLC2 mutants are not easy to work with as they display gametophyte lethality (Chen et al., 2019). A weak mutant allele exists but it is not in the same genetic background as the PIP biosensors and displays weak phenotypes (Kanehara et al., 2015). A strong allele is in Col-0 background but displays strong gametophyte lethality, the homozygote seedlings can grow but no seeds can be rescued from them (Chen et al., 2019). Still, we introduced a PIP biosensor into this genetic background and screened a high number of seedlings in the segregating population (that contains heterozygotes, wild-type and a low proportion of homozygous mutants) to select homozygote seedlings expressing the biosensor. In this background, we did not find 25% of homozygotes but rather around 5% which confirmed the gametophyte lethality of pi-plc2 mutant. Using a PI4P biosensor, our results show that the PI4P level increases at the TGN as well as at the PM in the pi-plc2 mutant background, consistently with our findings on the localization of PLC2. These results add a genetic confirmation to our findings.

Minor points that the author may want to address:

The depiction of PI-PLCs localization at the TGN and its mis-localization after metazachlor would have been informative to support the proteomic approach.

Using cell fractionation, previous studies have suggested that PI-PLCs could locate not only at the PM but as well in endomembrane compartments (Otterhag et al., 2001). We had a careful look at PI-

PLC2-venus construct driven by its own promoter (Kanehara et al., 2015) but could not detect any intracellular compartments labelled. However, the signal intensity in that line was extremely weak even at the PM. Thus, we decided to generate a PI-PLC2-GFP line driven by a stronger promoter, the Ubiquitin10 promoter which is used in plants to get a reasonable good level of expression (Geldner et al., 2009). Our pUBQ10::PLC2-GFP line showed a strong labeling at PM but also a weak labeling in intracellular structures. The difference of intensity between PM and TGN complicated the imaging and quantification. To tackle this issue, we had to apply a Gaussian filter and subtract the background, in these conditions we could detect intracellular structures that partly co-localized with a TGN marker. This result confirmed that PI-PLC2 localizes at the TGN additionally to the PM. Unfortunately, due to the low signal of PI-PLC2 at the TGN we could not quantify the intensity of fluorescence at the TGN in either control or metazachlor-treated seedlings. However, we are confident that PI-PLCs impact PI4P homeostasis at TGN given that we quantified an increase of PI4P at the TGN upon PI-PLC inhibitor or in the pi-plc2 knock-out mutant. Our data with the PI-PLC inhibitor revealed the PI4P increase at TGN cannot be increased by addition of metazachlor, suggesting that VLCFAs of SLs and PI-PLCs act in one pathway.

mCIT-1xPHFAPP1-E50A and mCIT-1xPHFAPP1-E50A-H54A intensity at the PM were not decreased upon metazachlor, but mCIT-1xPHFAPP1 sensor intensity at the PM was sensitive. Could the author, please, better explain this difference.

The 1xPH^{FAPP1} biosensor is binding to PI4P as well as ARF1, a small GTPase that localizes at the TGN. If the effect of metazachlor on the PM PI4P pool are weak, we expect 1xPH^{FAPP1} to be more sensitive to this difference because of the coincident detection of both PI4P and ARF1, which will drag this sensor toward the TGN (Noack et al., 2020). We have added a discussion about this in our revised manuscript.

The authors immunopurified SYP61-SVs/TGN compartments (Fig. 2d) and characterized the fatty acid (FA) and sterol composition by GC-MS in untreated and metazachlor-treated seedlings and did not detect significant alterations in sterol content. I am not sure if this negative result strongly argues against a sterol exchange mechanism. Besides technical/sensitivity issues, there could be a mechanism in place that balances the sterol concentrations after the exchange mechanism. The discussion could be more balanced here.

We agree. We modified the text to be more cautious in our interpretations.

Would available pi4 kinase mutants have altered sensitivity when grown on metazachlor?

To address this question we used the PI4K β 1; β 2 double mutant as PI4K β 1 has been shown to localize at the TGN and acts redundantly with PI4K β 2 (Kang et al., 2011; Preuss et al., 2006). Actually, the PI4K β 1; β 2 double mutant displays swollen TGN, similarly to metazachlor-treatment (Kang et al., 2011). Moreover, mis-localization of PIN2 has also been described in the PI4K β 1; β 2 double mutant (Lin et al., 2019). As a phenotypic readout of metazachlor-treatment and PIN2 mis-sorting is the root gravitropism, we analyzed root gravitropic response of the PI4K β 1; β 2 double mutant in untreated and metazachlor-treated seedlings, as compared to wild-type. Our results show that while the wild-type displays altered gravitropic response upon metazachlor, the PI4K β 1; β 2 double mutant is relatively more resistant or insensitive to the treatment. These results suggest that the pool of PI4P synthesized at the TGN is required to mediate the effect of metazachlor, which is consistent and reinforce our data by genetic mean.

References used in this letter

- Chen, X., Li, L., Xu, B., Zhao, S., Lu, P., He, Y., Ye, T., Feng, Y.-Q., Wu, Y., 2019. Phosphatidylinositol-specific phospholipase C2 functions in auxin-modulated root development. *Plant. Cell Environ.* 42, 1441–1457. <https://doi.org/10.1111/pce.13492>
- Clark, J., Anderson, K.E., Juvin, V., Smith, T.S., Karpe, F., Wakelam, M.J.O., Stephens, L.R., Hawkins, P.T., 2011. Quantification of PtdInsP3 molecular species in cells and tissues by mass spectrometry. *Nat. Methods* 8, 267–272. <https://doi.org/10.1038/nmeth.1564>
- Geldner, N., Déneraud-Tendon, V., Hyman, D.L., Mayer, U., Stierhof, Y.D., Chory, J., 2009. Rapid, combinatorial analysis of membrane compartments in intact plants with a multicolor marker set. *Plant J* 59, 169–178. <https://doi.org/10.1111/j.1365-313X.2009.03851.x>
- Heilmann, M., Heilmann, I., 2013. Mass measurement of polyphosphoinositides by thin-layer and gas chromatography. *Methods Mol. Biol.* 1009, 25–32. https://doi.org/10.1007/978-1-62703-401-2_3
- Joubès, J., Raffaele, S., Bourdenx, B., Garcia, C., Laroche-Traineau, J., Moreau, P., Domergue, F., Lessire, R., 2008. The VLCFA elongase gene family in *Arabidopsis thaliana*: Phylogenetic analysis, 3D modelling and expression profiling. *Plant Mol. Biol.* 67, 547–566. <https://doi.org/10.1007/s11103-008-9339-z>
- Kanehara, K., Yu, C.-Y., Cho, Y., Cheong, W.-F., Torta, F., Shui, G., Wenk, M.R., Nakamura, Y., 2015. *Arabidopsis* AtPLC2 Is a Primary Phosphoinositide-Specific Phospholipase C in Phosphoinositide Metabolism and the Endoplasmic Reticulum Stress Response. *PLoS Genet.* 11, e1005511. <https://doi.org/10.1371/journal.pgen.1005511>
- Kang, B.-H., Nielsen, E., Preuss, M.L., Mastronarde, D., Staehelin, L.A., 2011. Electron Tomography of RabA4b- and PI-4K β 1-Labeled Trans Golgi Network Compartments in *Arabidopsis*. *Traffic* 12, 313–329. <https://doi.org/10.1111/j.1600-0854.2010.01146.x>
- Kim, J., Jung, J.H., Lee, S.B., Go, Y.S., Kim, H.J., Cahoon, R., Markham, J.E., Cahoon, E.B., Suh, M.C., 2013. *Arabidopsis* 3-ketoacyl-coenzyme a synthase9 is involved in the synthesis of tetracosanoic acids as precursors of cuticular waxes, suberins, sphingolipids, and phospholipids. *Plant Physiol.* 162, 567–580. <https://doi.org/10.1104/pp.112.210450>
- Kleine-Vehn, J., Łangowski, Ł., Wiśniewska, J., Dhonukshe, P., Brewer, P.B., Friml, J., 2008. Cellular and molecular requirements for polar PIN targeting and transcytosis in plants. *Mol. Plant* 1, 1056–1066. <https://doi.org/10.1093/mp/ssn062>
- Lee, S.B., Jung, S.J., Go, Y.S., Kim, H.U., Kim, J.K., Cho, H.J., Park, O.K., Suh, M.C., 2009. Two *Arabidopsis* 3-ketoacyl CoA synthase genes, KCS20 and KCS2/DAISY, are functionally redundant in cuticular wax and root suberin biosynthesis, but differentially controlled by osmotic stress. *Plant J.* 60, 462–475. <https://doi.org/10.1111/j.1365-313X.2009.03973.x>
- Lin, F., Krishnamoorthy, P., Schubert, V., Hause, G., Heilmann, M., Heilmann, I., 2019. A dual role for cell plate-associated PI4K β in endocytosis and phragmoplast dynamics during plant somatic cytokinesis. *EMBO J.* 38. <https://doi.org/10.15252/embj.2018100303>
- Mamode Cassim, A., Grison, M., Ito, Y., Simon-Plas, F., Mongrand, S., Boutté, Y., 2020. Sphingolipids in plants: a guidebook on their function in membrane architecture, cellular processes, and environmental or developmental responses. *FEBS Lett.* <https://doi.org/10.1002/1873-3468.13987>
- Markham, J.E., Molino, D., Gissot, L., Bellec, Y., Hématy, K., Marion, J., Belcram, K., Palauqui, J.-C., Siatat-Jeunemaître, B., Faure, J.-D., 2011. Sphingolipids containing very-long-chain fatty acids define a secretory pathway for specific polar plasma membrane protein targeting in *Arabidopsis*. *Plant Cell* 23, 2362–78. <https://doi.org/10.1105/tpc.110.080473>
- Murata, N., Sato, N., Takahashi, N., 1984. Very-long-chain saturated fatty acids in phosphatidylserine from higher plant tissues. *Biochim. Biophys. Acta (BBA)/Lipids Lipid Metab.* 795, 147–150.

[https://doi.org/10.1016/0005-2760\(84\)90115-2](https://doi.org/10.1016/0005-2760(84)90115-2)

- Noack, L.C., Bayle, V., Armengot, L., Rozier, F., Mamode-Cassim, A., Stevens, F.D., Caillaud, M.C., Munnik, T., Mongrand, S., Jaillais, Y., 2020. A nanodomain anchored-scaffolding complex is required for PI4K α function and localization in plants. *bioRxiv*. <https://doi.org/10.1101/2020.12.08.415711>
- Otterhag, L., Sommarin, M., Pical, C., 2001. N-terminal EF-hand-like domain is required for phosphoinositide-specific phospholipase C activity in *Arabidopsis thaliana*. *FEBS Lett.* 497, 165–170. [https://doi.org/10.1016/S0014-5793\(01\)02453-X](https://doi.org/10.1016/S0014-5793(01)02453-X)
- Platre, M.P., Noack, L.C., Doumane, M., Bayle, V., Simon, M.L.A., Maneta-Peyret, L., Fouillen, L., Stanislas, T., Armengot, L., Pejchar, P., Caillaud, M.-C., Potocký, M., Čopič, A., Moreau, P., Jaillais, Y., 2018. A Combinatorial Lipid Code Shapes the Electrostatic Landscape of Plant Endomembranes. *Dev. Cell* 45, 465-480.e11. <https://doi.org/10.1016/j.devcel.2018.04.011>
- Pokotylo, I., Kolesnikov, Y., Kravets, V., Zachowski, A., Ruelland, E., 2014. Plant phosphoinositide-dependent phospholipases C: Variations around a canonical theme. *Biochimie*. <https://doi.org/10.1016/j.biochi.2013.07.004>
- Preuss, M.L., Schmitz, A.J., Thole, J.M., Bonner, H.K.S., Otegui, M.S., Nielsen, E., 2006. A role for the RabA4b effector protein PI-4K β 1 in polarized expansion of root hair cells in *Arabidopsis thaliana*. *J. Cell Biol.* 172, 991–998. <https://doi.org/10.1083/JCB.200508116>
- Qin, Y.M., Hu, C.Y., Pang, Y., Kastaniotis, A.J., Hiltunen, J.K., Zhu, Y.X., 2007. Saturated very-long-chain fatty acids promote cotton fiber and *Arabidopsis* cell elongation by activating ethylene biosynthesis. *Plant Cell* 19, 3692–3704. <https://doi.org/10.1105/tpc.107.054437>
- Tresch, S., Heilmann, M., Christiansen, N., Looser, R., Grossmann, K., 2012. Inhibition of saturated very-long-chain fatty acid biosynthesis by mefluidide and perfluidone, selective inhibitors of 3-ketoacyl-CoA synthases. *Phytochemistry* 76, 162–171. <https://doi.org/10.1016/j.phytochem.2011.12.023>
- Wang, W., Yang, X., Tangchaiburana, S., Ndeh, R., Markham, J.E., Tsegaye, Y., Dunn, T.M., Wang, G.-L., Bellizzi, M., Parsons, J.F., Morrissey, D., Bravo, J.E., Lynch, D. V, Xiao, S., 2008. An inositolphosphorylceramide synthase is involved in regulation of plant programmed cell death associated with defense in *Arabidopsis*. *Plant Cell* 20, 3163–79. <https://doi.org/10.1105/tpc.108.060053>
- Wattelet-Boyer, V., Brocard, L., Jonsson, K., Esnay, N., Joubès, J., Domergue, F., Mongrand, S., Raikhel, N., Bhalerao, R.P., Moreau, P., Boutté, Y., 2016. Enrichment of hydroxylated C24- and C26-acyl-chain sphingolipids mediates PIN2 apical sorting at trans-Golgi network subdomains. *Nat. Commun.* 7, 12788. <https://doi.org/10.1038/ncomms12788>

REVIEWERS' COMMENTS

Reviewer #1 (Remarks to the Author):

I am satisfied that the authors have done the best they could to answer to the critical points that were brought up in my review of the original version of this manuscript. The conclusions are now better supported by the data and I recommend acceptance. Howard Riezman

Reviewer #2 (Remarks to the Author):

The authors have fully addressed my comments and I believe the paper is now suitable for publication in NatComm.